# Human *BRCA* pathogenic variants were originated during recent human history

Jiaheng Li, Bojin Zhao, Teng Huang, Zixin Qin, San Ming Wang

*BRCA1* and *BRCA2* (*BRCA*) play essential roles in maintaining genome stability. *BRCA* germline pathogenic variants increase cancer risk. However, the evolutionary origin of human *BRCA* pathogenic variants remains largely elusive. We tested the 2,972 human *BRCA1* and 3,652 human *BRCA2* pathogenic variants from ClinVar database in 100 vertebrates across eight clades, but failed to find evidence to show cross-species evolution conservation as the origin; we searched the variants in 2,792 ancient human genome data, and identified 28 *BRCA1* and 22 *BRCA2* pathogenic variants in 44 cases dated from 45,000 to 300 yr ago; we analyzed the haplotype-dated human *BRCA* pathogenic founder variants, and observed that they were mostly arisen within the past 3,000 yr; we traced ethnic distribution of human *BRCA* pathogenic variants, and found that the majority were present in single or a few ethnic populations. Based on the data, we propose that human *BRCA* pathogenic variants were highly likely arisen in recent human history after the latest out-of-Africa migration, and the expansion of modern human population could largely increase the variation spectrum.

## Introduction

Repairing the damaged DNA by environmental and metabolic factors is vital for all lives on earth. In eukaryotes, this is achieved by the DNA damage repair system composed of multiple pathways to repair different types of DNA damage (Jeggo et al, 2016). The homologous recombination pathway repairs double-strand DNA damage by joint activities of a group of genes to reach error-free repair of DNA double-strand break (Jasin & Rothstein, 2013). *BRCA1* and *BRCA2* (*BRCA*) are two of the essential genes in the homologous recombination pathway. *BRCA1* was arisen in animal and plant kingdoms 1.2 billion yr ago, and *BRCA2* was arisen in fungus, plant, and animal kingdoms 1.6 billion yr ago (Kojic et al, 2002; Pfeffer et al, 2017). Studies revealed that *BRCA* across wide-range species including most of the primates is evolutionarily highly conserved by

negative selection in reflecting its essential roles in maintaining genome stability. In contrast, however, *BRCA* in humans and its closest living relatives of chimps and bonobos is rapidly evolving by positive selection (Huttley et al, 2000; Fleming et al, 2003; Abkevich et al, 2004; Pavlicek et al, 2004; Burk-Herrick et al, 2006; O'Connell, 2010; Lou et al, 2014). The positive selection likely resulted in the new function of *BRCA* gained in these three species, such as promoting immunity to counter viral infection (Lou et al, 2014), regulating gene expression and metabolism (Rosen et al, 2006; Chen et al, 2020), enhancing neural development (Pao et al, 2014), and increasing reproduction (Smith et al, 2013).

*BRCA* is one of the best-known genetic predisposition genes for cancer (Narod & Foulkes, 2004). Efforts made in the past decades have identified nearly 70,000 human *BRCA* variants (Cline et al, 2018) (https://brcaexchange.org/factsheet). A part of human *BRCA* variants is determined as "pathogenic" or "likely pathogenic" (*BRCA* PLP) in causing high cancer risk affecting mostly breast and ovary (Welcsh & King, 2001). *BRCA* PLP is widely used in clinical practice as the marker for cancer diagnosis, prevention, prognosis, and treatment through synthetic lethal mechanism (Anderson et al, 2008; Plon et al, 2008; Huen et al, 2010; George et al, 2017; Bhaskaran et al, 2019).

Human *BRCA* PLP provides an ideal system to study evolution origin of human disease susceptibility. While high evolutionary conservation across species suggests the possibility that human *BRCA* PLP could be originated from evolution conservation, the positive selection imposed in the humans, chimps, and bonobos but not in other species highlights another possibility that human *BRCA* PLP could be originated from human itself rather than from evolution conservation. However, there is no consent so far in determining which of the two possibilities could be the right origin for human *BRCA* PLP. We analyzed human *BRCA* variation data reported by previous evolutionary studies in *BRCA* (Huttley et al, 2000; Fleming et al, 2003; Burk-Herrick et al, 2006; Lou et al, 2014). The results showed that of the 111 human *BRCA* variants analyzed in these studies, 108 (94.7%) were non-pathogenic including 42 (37.8%) benign, 47 (42.3%) variants of uncertain significance (VUS), but only 6 (5.4%) were Pathogenic (Table S1). Therefore, the information from the previous studies reflects mainly evolutionary conservation of *BRCA* variation but not the origin of human *BRCA* PLP.

MoE Frontiers Science Center for Precision Oncology, Cancer Center and Institute of Translational Medicine, Faculty of Health Sciences, University of Macau, Macau, China

Correspondence: sanmingwang@um.edu.mo

In the current study, we addressed the evolution origin of human *BRCA* PLP. The rapid progress of genomics studies provides rich DNA sequence data across a wide range of species for phylogenic study, and the recent anthropological studies have also generated abundant DNA sequence data from ancient humans. These rich resources provide unique opportunities to study the evolutionary origin of the human *BRCA* PLP on the scale and accuracy unimaginable before. Taking the advantages, we performed a comprehensive phylogenic and archeological study to investigate the evolution origin of human *BRCA* PLP. Data from our study provide evidence to show that human *BRCA* PLP was highly unlikely originated from cross-species evolutionary conservation, but most likely arisen during recent human history after the latest out-of-Africa migration and the great expansion of modern human population.

## Results

### Phylogenetic analysis of human *BRCA* PLPs in non-human vertebrates

We identified a total of 6,624 *BRCA* PLP variants (2,972 in *BRCA1* and 3,652 in *BRCA2*) from the ClinVar database for the study (Tables 1 and S2). We searched evidence for potential conservation of *BRCA* PLP variants between the humans and the 100 vertebrates distributed in eight clades of Primate, Euarchontoglires, Laurasiatheria, Afrotheria, Mammal, Aves, Sarcopterygii, and Fish. We identified 172 (5.8%) human *BRCA1* PLP variants shared with 69 species, and 312 (8.6%) human *BRCA2* PLP variants shared with 90 species (Figs 1, 2, and S1A and Tables S3 and S4).

Of the *BRCA1* PLP variants shared in the eight clades, Aves had the highest sharing number of 14 species and Sarcopterygii the second highest of eight species than other clades (P < 0.001) (Fig S2). For example, *BRCA1* c.3268C>T (p.Gln1090Ter) was shared with 19 species from Rock pigeon in Aves to Lizard in Sarcopterygii, c.2498T>A (p.Leu833Ter) with 18 species from Saker falcon in Aves to Spiny softshell turtle in Sarcopterygii, and c.2138C>A (p.Ser713Ter) with 18 species from Rock pigeon in Aves to Lizard in Sarcopterygii; all 11 PLP variants located in the BRCA1 BRCT domain were shared with the species within Aves and Sarcopterygii clades. Tasmanian devil, a species with high risk of developing facial cancer, also shared 20 human *BRCA1* PLP variants. In the shared *BRCA2* PLP variants, the highest one including *BRCA2* c.8933C>A (p.Ser2978Ter) shared with 36 species from Lesser Egyptian jerboa in Euarchontoglires to Opossum in mammal, and c.8933C>A (p.Ser2978Ter) shared with 36 species from Lesser Egyptian jerboa in Euarchontoglires to Opossum in Mammal to Spotted gar in Fish, the most distal species sharing this PLP. As the most used animal models in biomedical study, mouse shared eight and rat shared seven human *BRCA1* PLP variants, of which only two were located at the BRCT domain. Zebrafish, another important biological model, shared 25 human *BRCA2* PLP variants.

Of the species sharing with human *BRCA* PLP variants, Spiny softshell turtle in Sarcopterygii had the highest sharing number of 34 *BRCA1* PLP variants, and Lizard in Sarcopterygii had the highest sharing number of 57 *BRCA2* PLP variants. There were no human *BRCA* PLP variant shared with primates of Chimp, Gorilla, Orangutan, Gibbon, Rhesus, Crab-eating macaque and Baboon whereas there were 10 PLP variants shared with the distal primates of Marmoset, Squirrel monkey, and Bushbaby: Marmoset shared *BRCA1* c.850C>T (p.Gln284Ter) and *BRCA2* c.1642C>T (p.Gln548Ter), Squirrel monkey and Bushbaby shared an *BRCA1* intronic c.1058G>A (p.Trp353Ter), Squirrel monkey and Bushbaby shared *BRCA2* c.4689G>A (p.Trp1563Ter), and Bushbaby shared *BRCA2* c.2651C>A (p.Ser884Ter), c.2978G>A (p.Trp993Ter), c.4689G>A (p.Trp1563Ter), c.3504G>T (p.Met1168Ile) and c.5263G>T (p.Glu1755Ter). Marmoset, Squirrel monkey, and Bushbaby had the divergent time of 40, 12.5, 15 million yr from hominins, accordingly. As control, we searched *BRCA1* c. 68_69del, *BRCA1* c.5266dup and *BRCA2* c.5946del, the three *BRCA* founder PLP variants in Ashkenazi Jews population (Levy-Lahad et al, 1997), in the 100 vertebrates. We found no evidence for their presence in these species: the wildtype AG at the position of 68–69 in *BRCA1* was present across 51 species from Chimp to Armadillo, the wild-type C at the position of 5,266 in *BRCA1* was present across 83 species from Rhesus to Coelacanth, and the wild-type T at the position of 5,946 of *BRCA2* was present across 87 species from Chimp to Tetraodon.

Of the 172 *BRCA1* and 312 *BRCA2* PLP variants shared with non-human species, the major types were stop-gain/nonsense variants (156 [90.1%] shared *BRCA1* PLP variants and 280 [89%] shared *BRCA2* PLP variants). Although frameshift indels constituted the majority of human *BRCA* PLP variants, none of them was present in other species (Table 1). The shared PLP variants did not enrich in specific functional domains, but distributed across coding region in both *BRCA1* and *BRCA2* (Fig S3A and B, P > 0.1): of the 388 of 2,972 (13%) *BRCA1* PLP variants located in functional domains, only 13 (3.4%) were shared with other species (11 in BRCT domain and 2 in Coiled-Coil domain); of the 1,077 of 3,652 (29.5%) *BRCA2* PLP variants in functional domains, only 54 (5.0%) were shared with other species (15 in BRC repeats, 38 in DBD, 1 in NLS and 8 in Transactivation domain) (Tables S3 and S4).

The results from our phylogenetic analysis reported above do not support evolution conservation as the major source of human *BRCA* PLP.

### Archeological analysis of *BRCA* PLP in ancient humans

As our phylogenetic analysis did not find evidence to support evolution conservation as the major source of human *BRCA* PLP, we then tested whether human *BRCA* PLP could be originated from human itself. We performed an archeological analysis by searching human *BRCA* PLP in ancient human genome sequence data. We first searched the three Neanderthal and two Denisovan genome sequences (Green et al, 2010; Meyer et al, 2012; Prufer et al, 2014, 2017; Mafessoni et al, 2020), which were diverged from the ancestors of modern humans 400–800 thousand year ago (kya). We did not identify any matched human *BRCA* PLP variants. Next, we searched human *BRCA* PLP variants in 2,792 ancient human genome sequences dated from 45,000 to 300 yr ago. We identified 46 *BRCA* PLP variants in 50 ancient individuals, including 24 *BRCA1* PLP variants in 28 individuals and 22 *BRCA2* PLP variants in 22 individuals (Table 2 and Fig S1B), lived in Africa, Europe, Asia, Oceania, and Central and North America (Fig 3). Each matched PLP variant had dbSNP ID, further ensured the fidelity of the matched variants. The oldest matched *BRCA1* PLP variant was *BRCA1* c.181T>G (p.C61G) in an individual in Voronezh Oblast, Russia dated about 37,470 yr ago (Seguin-Orlando et al, 2014), the youngest matched *BRCA1* PLP

**Table 1.  Human *BRCA* PLP variants in 100 vertebrates.**

| | BRCA1 | BRCA2 | Total (%) |
|---|---|---|---|
| A. Human BRCA PLP used in the study | | | |
| Total | 2,972 | 3,652 | 6,624 |
| Pathogenic | 2,767 | 3,396 | 6,163 |
| Likely pathogenic | 110 | 155 | 265 |
| Pathogenic/likely pathogenic | 95 | 101 | 196 |
| Types of mutation | | | |
| Frameshift deletion | 1,250 | 1,676 | 2,926 (44.2) |
| Stopgain/nonsense | 655 | 851 | 1,506 (22.7) |
| Frameshift insertion | 622 | 738 | 1,360 (20.5) |
| Splice site | 176 | 190 | 366 (5.5) |
| Nonsynonymous SNV | 116 | 47 | 163 (2.5) |
| Frameshift substitution | 88 | 114 | 202 (3.1) |
| Intron variant | 48 | 18 | 66 (1.0) |
| Nonframeshift deletion | 10 | 12 | 22 (0.3) |
| Nonframeshift substitution | 6 | 5 | 11 (0.2) |
| 5'/3' UTR | 1 | 1 | 2 (0.03) |
| B. Human BRCA PLP shared with other species | | | |
| Shared mutation | 172 (5.8) | 312 (8.6) | 484 (7.3) |
| Types of shared mutation | | | |
| Stopgain/nonsense | 156 | 280 | 436 (6.6) |
| Splice site | 6 | 17 | 23 (0.3) |
| Nonsynonymous SNV | 5 | 8 | 13 (0.2) |
| Intron variant | 5 | 7 | 12 (0.2) |
| Frameshift deletion | — | — | — |
| Frameshift insertion | — | — | — |
| Nonframeshift deletion | — | — | — |
| 5'/3' UTR | — | — | — |
| C. Number of species sharing human BRCA PLP | | | |
| 1 | 68 (39.5) | 130 (41.6) | 198 (40.9) |
| 2 | 24 | 40 | 65 |
| 3 | 14 | 36 | 50 |
| 4 | 7 | 18 | 25 |
| 5 | 19 | 15 | 34 |
| 6 | 6 | 12 | 18 |
| 7 | 8 | 8 | 16 |
| 8 | 3 | 5 | 8 |
| 9 | 2 | 4 | 6 |
| 10 | 2 | 4 | 6 |
| 11 | 4 | 1 | 5 |
| 12 | 2 | 2 | 4 |
| 13 | 1 | 4 | 5 |
| 14 | 1 | 3 | 4 |
| 15 | 3 | 5 | 8 |

**Table 1.  Continued**

|  | BRCA1 | BRCA2 | Total (%) |
|---|---|---|---|
| 16 | 2 | 5 | 7 |
| 17 | 3 | 2 | 5 |
| 18 | 2 | 6 | 8 |
| 19 | 1 | 2 | 4 |
| 20 | — | 4 | 4 |
| 21 | — | 2 | 2 |
| 22 | — | — | — |
| 23 | — | — | — |
| 24 | — | — | — |
| 25 | — | 1 | 1 |
| 26–33 | — | — | — |
| 34 | — | 1 | 1 |
| 35 | — | — | — |
| 36 | — | 1 | 1 |
| 37–56 | — | — | — |
| 57 | — | 1 | 1 |
|  | 172 (100) | 312 (100) | 484 (100) |

variant was *BRCA1* c.2599C>T (Q867X) in an individual in Fujian, China dated 307 yr ago (Yang et al, 2020); the oldest matched *BRCA2* PLP variant was *BRCA2* c.9573G>A (p.W3191X) in an individual in Bor District, Serbia dated about 7,874 yr ago (Mathieson et al, 2018), the youngest matched *BRCA2* PLP variant was *BRCA2* c.8009C>T (p.S2670L) in an individual in Shefa, Vanuatu dated 350 yr ago (Lipson et al, 2020). Of the 24 matched *BRCA1* PLP variants, 14 (58.3%) were stop-gain and six (25%) were located at the BRCT domain; of the 22 matched *BRCA2* PLP variants, 15 (68.2%) were stop-gain and three were located at the nucleic acid binding domain. Four of the 24 *BRCA1* PLP variants were detected in two individuals.

### Dated *BRCA* founder PLP and ethnic-specific distribution of *BRCA* PLP

Many *BRCA* PLP variants are determined as *BRCA* founder PLP variants in different human ethnic populations, and their arisen times were determined by haplotyping analysis. For example, the arisen times for the three *BRCA* PLP founder variants (*BRCA1* c.68_69del, *BRCA1* c.5266dup, and *BRCA2* c.5946del) in Ashkenazi Jews population were determined as 1,720, 1,800, and 580 yr ago, respectively (Table 3). We identified 34 *BRCA* founder PLP variants including 22 in *BRCA1* and 12 in *BRCA2*. Their arisen times were from 3,225 to 140 yr ago (Table 3). For example, *BRCA1* c.3228_3229del (p.G1077fs) was the oldest arisen 3,225 yr ago in Tuscany, Italy, *BRCA1* c.1175_1214del (p.L345fs) was the youngest arisen 180 yr ago; *BRCA2* c.9026_9030del (p.T3009fs) was the oldest arisen 2,760 yr ago in Spanish, *BRCA2* c.9118-2A>G was the youngest arisen 220-144 yr ago in Finnish.

We also traced the ethnic origins for the *BRCA* PLP variants used in the study. Of the 1,054 *BRCA* PLP variants with available ethnic information, 548 (52%) were originated from single ethnic population, 327 (31%) were shared only between two ethnic populations. The rates were consistent in both *BRCA1* and *BRCA2* PLP variants (Table S5A–C).

## Discussion

Our study analyzed the evolutionary origin of *BRCA* PLP in modern human population. Our phylogenetic analysis across 100 vertebrates found no direct evidence for cross-species evolution conservation as the source for human *BRCA* PLP. Our archeological analysis in 2,792 ancient human individuals dated back to 45,000 yr ago identified 46 BRCA PLP variants of which 45 were arisen within the last 10,000 yr. Our analysis in the haplotyping-dated human *BRCA* founder PLP variants observed that nearly all were arisen within the last 3,000 yr. We further traced the ethnic origins of the *BRCA* PLP variants used in the study and observed that the majority were present in single or few ethnic populations. Based on these observations, we propose that human *BRCA* PLP was most likely arisen in recent human history, possibly within a few thousand years, after the latest human out-of-Africa migration and settlement at different global destinations. We consider that the positive selection on human *BRCA* could play a major role, and the population expansion could further increase the spectrum of human *BRCA* PLP in modern human population (Fig 4).

The prevalence of *BRCA* PLP is between 0.2% and 0.5% in modern human population. For example, the prevalence is 0.26% in Japanese population (Momozawa et al, 2018), 0.29% in Macau population (Qin et al, 2021), 0.38% in Chinese population (Dong et al, 2021), 0.38% in Mexican population (Fernandez-Lopez et al, 2019), 0.39% in Malaysian population (Wen et al, 2018), 0.53% in Taiwanese population (Chian et al, 2021), and 0.53% in Caucasian populations

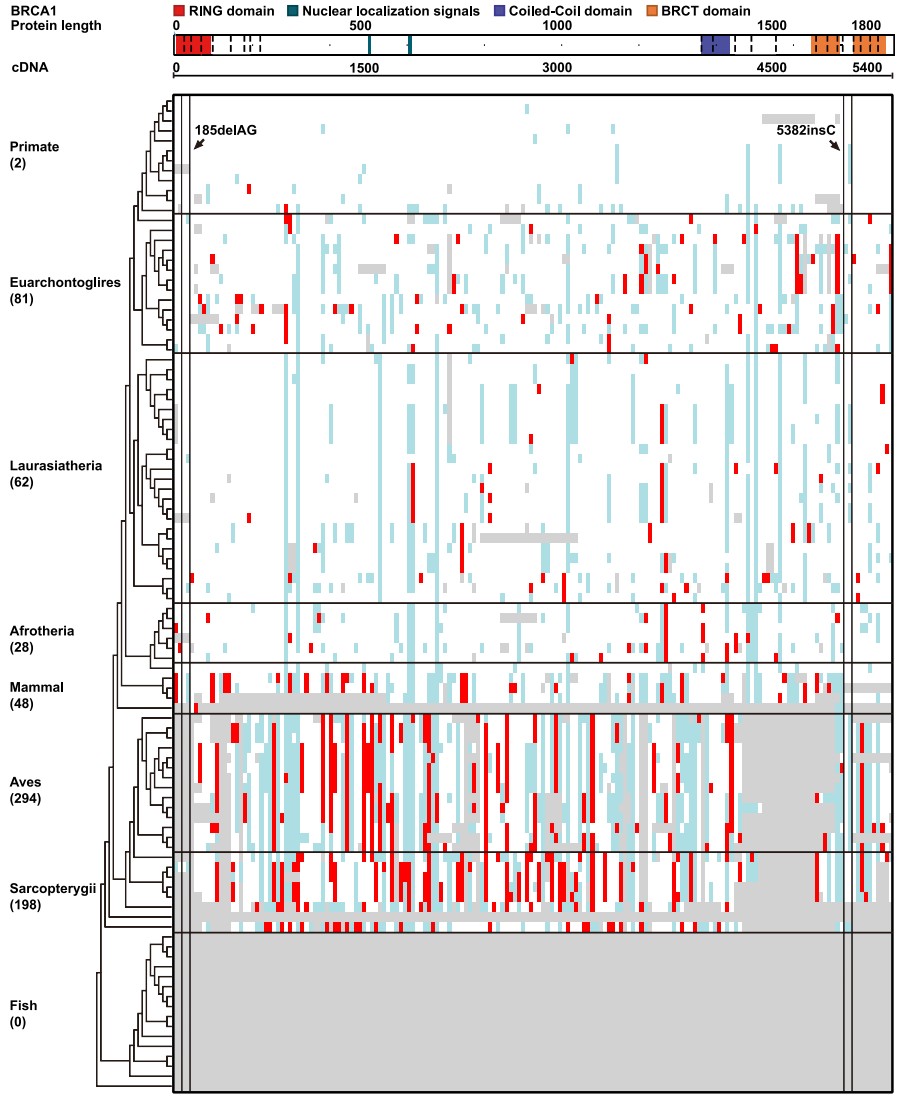

**Figure 1. Distribution of human *BRCA1* PLP variants in 100 vertebrates.**

Red cell: same as the human PLPs; empty cell: same base as human wild type; blue cell: different from human wild type and PLPs; gray cell: gaps; "-": no aligned base; "=": gap with at least one un-aligned base. The figure was generated using GraphPad Prism (version 9.0.0 for Windows, GraphPad Software). 185delAG, 5382insC: the *BRCA1* founder pathogenic variants in Ashkenazi Jews population.

(Kurian et al, 2019). That implies that one in several hundreds of human individuals carries a *BRCA* PLP variant. The prevalence of *BRCA* PLP can be the highest in disease-causing genetic predisposition genes in human. It is interesting to understand why *BRCA* PLP can reach such high level in human population regardless their deleterious impact. Possible explanations can be (1). The cancer caused by *BRCA* PLP occurs mostly at later reproduction stage. Before reaching the stage, the PLP has already been transmitted to the next generation; (2). The loss of tumor suppressing function of BRCA due to *BRCA* PLP could be developmental stage-dependent. It states that the PLP could be beneficial at the reproductive stage but deleterious at later reproduction stage. Positive selection imposed on the reproductive stage can select the *BRCA* PLP not deleterious at the stage. However, our current study does not determine whether one or both of the explanations could contribute to the higher prevalence of *BRCA* PLP in modern human population.

Extensive animal model studies, especially from mouse studies, have provided rich evidence showing the pathogenic consequences by BRCA PLP in non-human species. *Brca1* knockout-mice showed abnormal post-implantation development and embryonic proliferation (Liu et al, 1996; Hakem et al, 1996), embryonic lethality (Ludwig et al, 1997), neuroepithelial abnormalities (Gowen et al, 1996), irradiation hypersensitivity and genetic instability (Shen et al, 1998), abnormal T cell development (Mak et al, 2000; Xu et al, 2001), and tumorigenesis (Xu et al, 1999; Brodie et al, 2001). Without the presence of other mutated genes such as the mutated TP53, however, the mutated Brca1 alone is not sufficient to directly cause cancer (Xu et al, 2001). This can further explain that the higher prevalence of *BRCA* PLP in human may have less deleterious impact in the absence of synergetic effects from other genetic events.

Many human *BRCA* PLP variants were shared with species in Aves and Sarcopterygii clads. This could happen by chance rather than by cross-species conservation, as it is unlikely that the conservation would allow the presence of the gaps across such wide distances. Alternatively, it could also be related to species-specific pathogenicity of the same *BRCA* PLP variants that the *BRCA* PLPs in

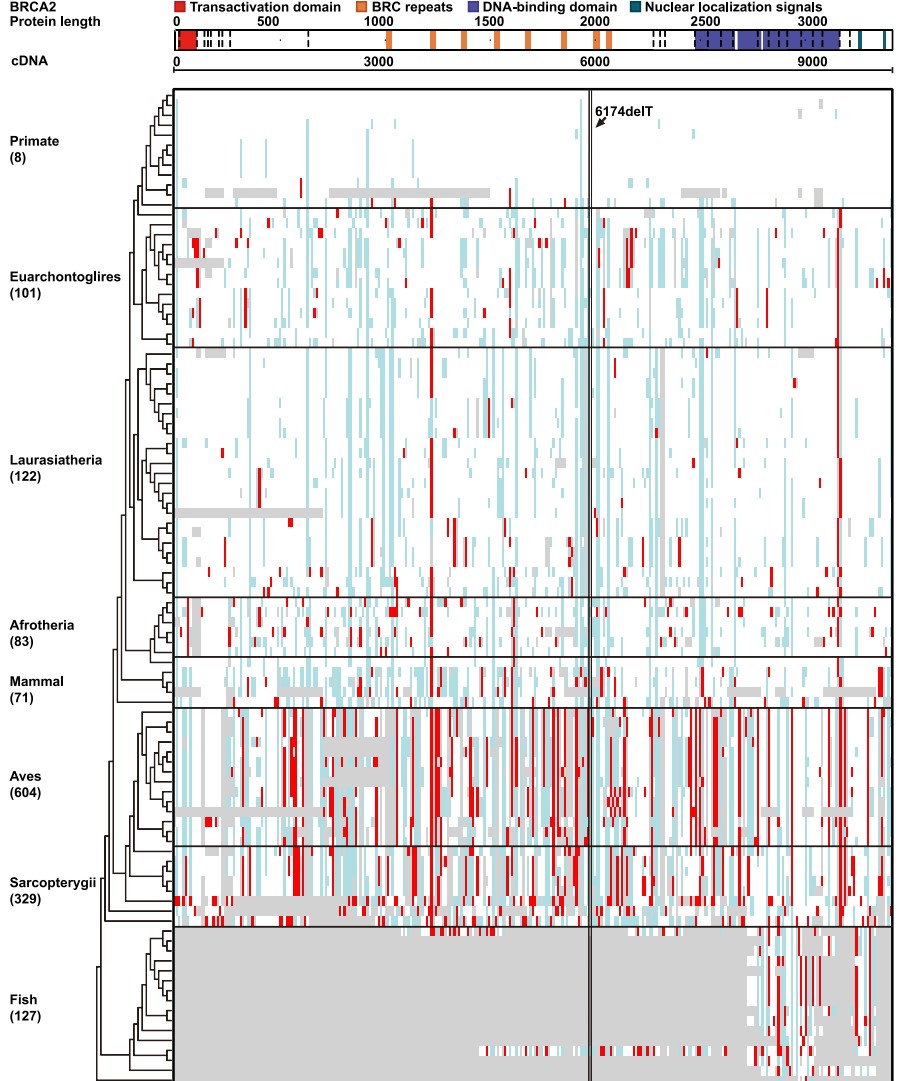

BRCA2 Protein length: Transactivation domain | BRC repeats | DNA-binding domain | Nuclear localization signals

0  500  1000  1500  2000  2500  3000

cDNA: 0  3000  6000  9000

6174delT

Primate (8)

Euarchontoglires (101)

Laurasiatheria (122)

Afrotheria (83)

Mammal (71)

Aves (604)

Sarcopterygii (329)

Fish (127)

**Figure 2.  Distribution of human *BRCA2* PLP variants in 100 vertebrates.**
Ibid as in Fig 1. 6174delT: the *BRCA2* founder pathogenic variants in Ashkenazi Jews population.

humans may not be pathogenic in non-human species (Gao & Zhang, 2003). For example, Tasmanian devil shared 20 human *BRCA* PLP variants. Tasmanian has a high risk of developing facial cancer, which is related with the fusion of chromosome 1 and X (Hawkins et al, 2006; Murchison et al, 2010; Taylor et al, 2017), but no evidence to show that the cancer is related to the *BRCA* PLP variants shared with human. A study analyzed human deleterious mutations in multiple genes including BRCA shared with mouse (Gao & Zhang, 2003), and evaluated multiple theories to explain the biological correlation including the "funder effect," "fixations of slightly deleterious mutations," "relaxed selection on late-onset phenotypes," and "compensatory changes." The study considered that the "compensatory changes," which stated that "compensatory mutations at other sites of the same or a different protein render the deleterious mutations neutral," can be the best to explain the conservation of human deleterious mutation with the distant species of mouse. We consider that the compensation theory can also be used to explain the sharing of

human *BRCA* PLPs with the species in Aves and Sarcopterygii clads, although direct evidence is not available to validate the explanation.

Only limited numbers of *BRCA* PLP variants were identified in the ancient humans. This can be related to the limited data available from ancient humans. In addition, genomic sequences in many ancient samples had limited coverage due to the rarity and poor quality of ancient DNA. Furthermore, the size differences between ancient and modern human populations can also be a factor. It is estimated that the number of indivuduals in the latest out-of-Africa migration 65,000–50,000 yr ago was about 1,000–2,500 (Henn et al, 2012), the size of the human population in 1,800s was about 1 billion, and the size of modern human population is close to eight billion (https://ourworldindata.org/world-population-growth, https://data.worldbank.org/indicator/SP.POP.TOTL). All human *BRCA* PLP variants used in our current study were derived from the current human population. With the positive selection, continually increased human population size, and more powerful DNA

**Table 2.** *BRCA* PLP variants identified in ancient humans.

| Age (BP) | Fossil site | Variation[a] | | Type | dbSNP155 | Domain | References[b] |
|---|---|---|---|---|---|---|---|
| | | cDNA | Protein | | | | |
| *BRCA1* | | | | | | | |
| 37,470 ± 1,210 | Voronezh Oblast, Russia | c.181T>G | p.C61G | missense | rs28897672 | Zinc finger | 38 |
| 9,871 ± 1,650 | Bor District, Serbia | **c.5095C>T** | **p.R1699W** | missense | rs55770810 | BRCT | 20 |
| 7,796.5 ± 75.5 | Shandong, China | c.2059C>T | p.Q687X | stopgain | rs273898674 | — | 35 |
| 4,971 ± 450 | Catalonia, Spain | c.1A>G | p.D2_M18del | startloss | rs80357287 | — | 27 |
| 4,671 ± 50 | Sistan and Baluchestan, Iran | **c.4327C>T** | **p.R1443X** | stopgain | rs41293455 | — | 24 |
| 4,121 ± 100 | Senec, Slovakia | **c.3607C>T** | **p.R1203X** | stopgain | rs62625308 | — | 31 |
| 3,350 ± 200 | Sardinia, Italy | c.1618G>T | p.E540X | stopgain | rs730881471 | — | 3 |
| 3,096 ± 20 | Hope Town, Bahamas | **c.5503C>T** | **p.R1835X** | stopgain | rs41293465 | BRCT | 4 |
| 2,110 ± 30 | Ballito Bay, South Africa | c.643C>T | p.Q215X | stopgain | rs886037979 | | 37 |
| 2,110 ± 30 | Ballito Bay, South Africa | c.949C>T | p.Q317X | stopgain | rs80357211 | | 37 |
| 2,110 ± 30 | Ballito Bay, South Africa | c.441+1G>A | — | splice site | rs397509172 | | 37 |
| 2,031 | Khovsgol, Mongolia | c.135-1G>A | — | splice site | rs80358158 | — | 2 |
| 1,980 ± 20 | Ballito Bay, South Africa | c.3967C>T | p.Q1323X | stopgain | rs80357262 | | 37 |
| 1,933 | Rostov Oblast, Russia | c.5353C>T | p.Q1785X | nonsense | rs80356969 | BRCT | 2 |
| 1,695 ± 195 | Chukotka, Russia | c.2761C>T | p.Q921X | nonsense | rs80357377 | — | 5 |
| 1,471 ± 50 | Girona, Catalonia | c.4255G>T | p.E1419X | stopgain | rs80357309 | — | 27 |
| 1,221 ± 100 | Monsenor Nouel, Dominica | c.2309C>A | p.S770X | stopgain | rs80357063 | — | 4 |
| 921 ± 500 | Monsenor Nouel, Dominica | **c.3607C>T** | **p.R1203X** | stopgain | rs62625308 | — | 4 |
| 921 ± 500 | Monsenor Nouel, Dominica | c.5096G>A | p.R1699Q | missense | rs41293459 | BRCT | 4 |
| 921 ± 500 | Monsenor Nouel, Dominica | **c.4327C>T** | **p.R1443X** | stopgain | rs41293455 | — | 4 |
| 921 ± 500 | Monsenor Nouel, Dominica | c.5216A>T | p.D1739V | missense | rs80357227 | BRCT | 4 |
| 921 ± 500 | Monsenor Nouel, Dominica | **c.5095C>T** | **p.R1699W** | missense | rs55770810 | BRCT | 4 |
| 721 ± 100 | San Pedro de Macors, Dominica | c.2389G>T | p.E797X | stopgain | rs62625306 | — | 4 |
| 721 ± 100 | San Pedro de Macors, Dominica | **c.5503C>T** | **p.R1835X** | stopgain | rs41293465 | BRCT | 4 |
| 480 ± 30 | Eland Cave, South Africa | c.5074G>A | p.D1692N | nonsynonymous | rs80187739 | BRCT | 37 |
| 480 ± 30 | Eland Cave, South Africa | c.3403C>T | p.Q1135X | stopgain | rs80357136 | | 37 |
| 419 ± 94 | Guangxi, China | c.4573C>T | p.Q1525X | stopgain | rs886040237 | SRD | 34 |
| 307.5 ± 26.5 | Fujian, China | c.2599C>T | p.Q867X | stopgain | rs886038001 | — | 35 |
| *BRCA2* | | | | | | | |
| 7,874 ± 72 | Bor District, Serbia | c.9573G>A | p.W3191X | stopgain | rs398122617 | NAB | 27 |
| 7,030 ± 50 | Castile and Leon, Spain | c.9466C>T | p.Q3156X | stopgain | rs276174925 | NAB | 27 |
| 4,950 ± 150 | Baden, Germany | c.6952C>T | p.R2318X | stopgain | rs80358920 | — | 14 |
| 4,571 ± 350 | Kladno, Central Bohemian Region | c.9382C>T | p.R3128X | stopgain | rs80359212 | — | 26 |
| 4,384 ± 200 | Scotland, UK | c.3922G>T | p.E1308X | stopgain | rs80358638 | — | 26 |
| 4,250 ± 1,200 | Vrancea County, Romania | c.7806-2A>G | — | splicing | rs81002836 | — | 9 |
| 3,858 ± 200 | Budapest, Hungary | c.8695C>T | p.Q2899X | stopgain | rs397507411 | — | 26 |
| 3,546 ± 325 | Atyrau region, Kazakhstan | c.244A>T | p.K82X | stopgain | rs397507628 | — | 24 |
| 2,921 ± 100 | NWFP, Pakistan | c.1189C>T | p.Q397X | stopgain | rs760815829 | — | 24 |
| 2,921 ± 100 | NWFP, Pakistan | c.3469G>T | p.E1157X | stopgain | rs80358595 | — | 24 |
| 2,821 ± 100 | East Kazakhstan, Kazakhstan | c.1825C>T | p.Q609X | stopgain | rs80358472 | — | 33 |
| 2,821 ± 100 | East Kazakhstan, Kazakhstan | c.2455C>T | p.Q819X | stopgain | rs397507629 | — | 33 |

**Table 2.** Continued

| Age (BP) | Fossil site | Variation[a] | | Type | dbSNP155 | Domain | References[b] |
|---|---|---|---|---|---|---|---|
| | | cDNA | Protein | | | | |
| 2,330 ± 25 | St. Helena, South Africa | c.2905C>T | p.Q969X | stopgain | rs886038080 | | 36 |
| 2,321 ± 100 | NWFP, Pakistan | c.5200G>T | p.E1734X | stopgain | rs786202543 | — | 24 |
| 2,131 | Khovsgol, Mongolia | c.316+1G>A | — | splice site | rs397507303 | — | 2 |
| 2,083 | Omnogovi, Mongolia | c.7617+1G>A | — | splice site | rs397507922 | — | 2 |
| 1,980 ± 20 | Ballito Bay, South Africa | c.7480C>T | p.R2494X | stopgain | rs80358972 | | 37 |
| 1,783 | Issyk Kul, Kyrgyzstan | c.7007G>A | p.R2336H | missense | rs28897743 | — | 2 |
| 1,697 | South Kazakhstan, Kazakhstan | c.7977-1G>A | — | splice site | rs81002874 | — | 2 |
| 1,100 ± 500 | Santo Domingo, Dominica | c.171C>A | p.Y57X | stopgain | rs201523522 | — | 4 |
| 480 ± 30 | Eland Cave, South Africa | c.475G>A | p.V159M | nonsynonymous | rs80358702 | | 37 |
| 350 ± 100 | Shefa, Vanuatu | c.8009C>T | p.S2670L | missense | rs80359035 | NAB | 18 |

[a]Bold refers to the variants detected in more than one individual.
[b]References listed in Table S6

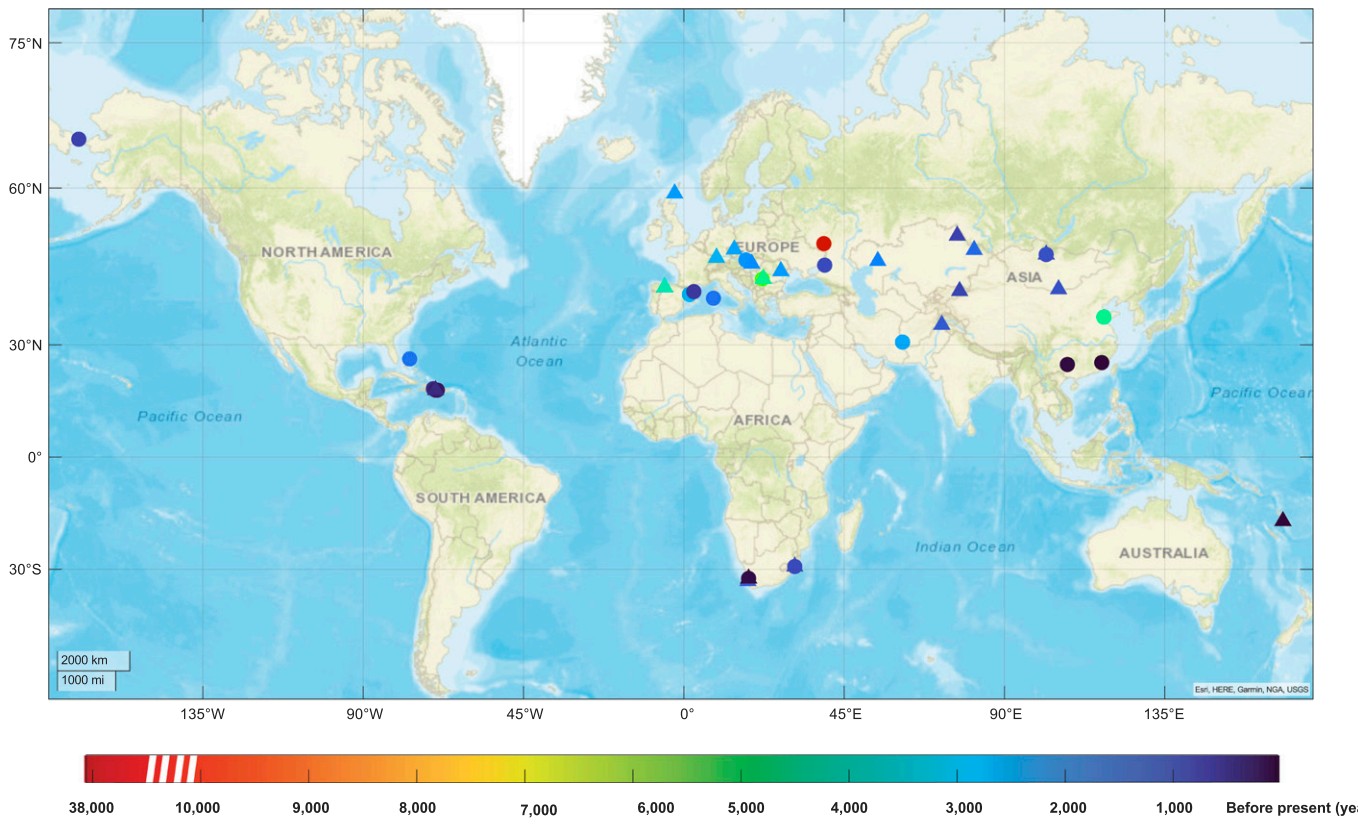

**Figure 3. Geographic distribution of human *BRCA* PLPs in ancient humans.**
Circle dots: matched *BRCA1* PLPs; triangle dot: matched *BRCA2* PLPs. The irregular dots: overlapped *BRCA1* and *BRCA2* PLPs. Colors in icons: time before present. The red circle dot was *BRCA1* c.181T>G, dated 37,470 ± 1,210 yr ago (Table 2).

sequencing technologies under developing, it is expected that more new human *BRCA* PLP variants would arise and be identified.

Human *BRCA1* has 1,863 residues and *BRCA2* has 3,418 residues. Although gene with large size can have more chance to generate more genetic variants, it is unlikely that the size in *BRCA* is a major

factor for the large quantity of human *BRCA* PLP. For example, our study showed that mouse *Brca* does not change although the size of mouse *Brca* is very close to the human BRCA (*Brca1* 1,812 residues and *Brca2* 3,329 residues) (Wang & Wang, 2021). Another factor related with ethnic-specific PLPs can be that the same *BRCA* PLP

**Table 3.** *BRCA* founder PLP variants dated by haplotype analysis.

| Age (BP) | Population | Mutation (HGVS) | | | References[a] |
|---|---|---|---|---|---|
| | | cDNA | Protein | Variation type | |
| *BRCA1* | | | | | |
| 3,225 | Tuscany, Italy | 3347delAG (c.3228_3229del) | p.G1077fs | frameshift | 62 |
| 2,400–1,600 | Iberia | c.3331_3334delCAAG | p.Q1064fs | frameshift | 39 |
| 1,800 | Ashkenazi Jewish | 5832insC (c.5266dupC) | p.Q1756fs | frameshift | 40 |
| 1,720 | Northeastern Italy | c.676delT | p.C226fs | frameshift | 41 |
| 1,500–750 | Ashkenazi Jewish | 185delAG (c.68_69del) | p.E23fs | frameshift | 21 |
| 1,500 | Swedish | 3171ins5 (c.3052_3053ins5) | — | — | 43 |
| 1,480 | Columbia | 3450delCAAG (c.3331_3334del) | p.Q1064fs | frameshift | 44 |
| 1,440 | Mexican | c.548-?_4185 + ?del | — | exon 9–12del | 45 |
| 800 | Moroccan | c.5309G>T | p.G1770V | missense | 46 |
| 750 | Tuscany, Italy | 1499insA (c,1380dup) | p.F414fs | frameshift | 47 |
| 720–460 | Finnish | 3744delT (c.3626del) | — | nonsense | 48 |
| 600 | Norwegians | 1675delA (c.1556del) | p.K472fs | frameshift | 49 |
| 600 | Norwegians | 1135insA (c.1135_1136insA) | p.K472fs | frameshift | 49 |
| 500 | Norwegians | 816delGT (c.697_698del) | p.V186fs | frameshift | 50 |
| 500 | Norwegians | 3347delAG (c.3228_3229del) | p.G1077fs | frameshift | 50 |
| 500 | South Africans | c.2641G>T | p.E881X | nonsense | 51 |
| 380 | Spanish | c.5153-1G>A | — | splice site | 52 |
| 275 | Greek | G1738R (c.5212G>A) | p.G1738R | missense | 53 |
| 200 | Dutch | 2804delAA (c.2685_2686del) | p.P850fs | frameshift | 54 |
| 200 | Afro-Americans | 943ins10 (c.824_825ins10) | — | — | 55 |
| <200 | Finnish | 4216-2A>G (c.4097-2A>G) | — | splice site | 56 |
| 180 | Unknown | 1294del40 (c.1175_1214del) | p.L345fs | frameshift | 48 |
| *BRCA2* | | | | | |
| 2,760 | Spanish | 9254del5 (c.9026_9030del) | p.T3009fs | frameshift | 57 |
| 2,600–2,400 | Portugal | 156-157insAlu (c.1205T>C) | p.L402P | missense | 58 |
| 1,904 | Spanish | 5344delAATA (c.5116_5119del) | p.N1706fs | frameshift | 59 |
| 1,600 | US and Canda | c.3036_3039del | p.S1013fs | frameshift | 60 |
| 1,365 | Spanish | 9538delAA (c.9310_9311delAA) | p.L3104fs | frameshift | 59 |
| 1,200 | Spanish | 5374delTATG (c.5146_5149del) | p.T1716fs | frameshift | 52 |
| 580 | Ashkenazi Jewish | 6174delT (c.5946del) | p.S1982fs | frameshift | 60 |
| 500 | Icelaner | 995del5 (c.771_775del) | p.A257fs | frameshift | 61 |
| 400–200 | Finnish | 7708C>T (c.7480C>T) | p.R2494X | nonsense | 58 |
| 360 | US and France | 982del4 (c.755_758del) | p.D252fs | frameshift | 60 |
| 220–140 | Finnish | 8555T>G (c.8327T>G) | p.L2776X | nonsense | 56 |
| 220–140 | Finnish | c.9118-2A>G | — | splice site | 56 |

[a]references are listed in Table S6.

could arise by chance in different ethnic populations at different times. For example, the *BRCA1* 68-69del founder variant in Ashkenazi Jews is also present in other ethnic populations although at very low frequency, and *BRCA1* 3607C>T shared between the Slovakia case dated 4,121 ± 100 ago and Dominica case dated 921 ± 500 ago (Table 2). However, this unlikely played a major role in contributing to human *BRCA* PLP. Pleiotropy effects may also exist that variation in non-*BRCA* genes may contribute to the selection of *BRCA* PLPs, particularly for these selected in the post-reproductive age (Williams, 1957). This is particularly interesting as *BRCA* PLP

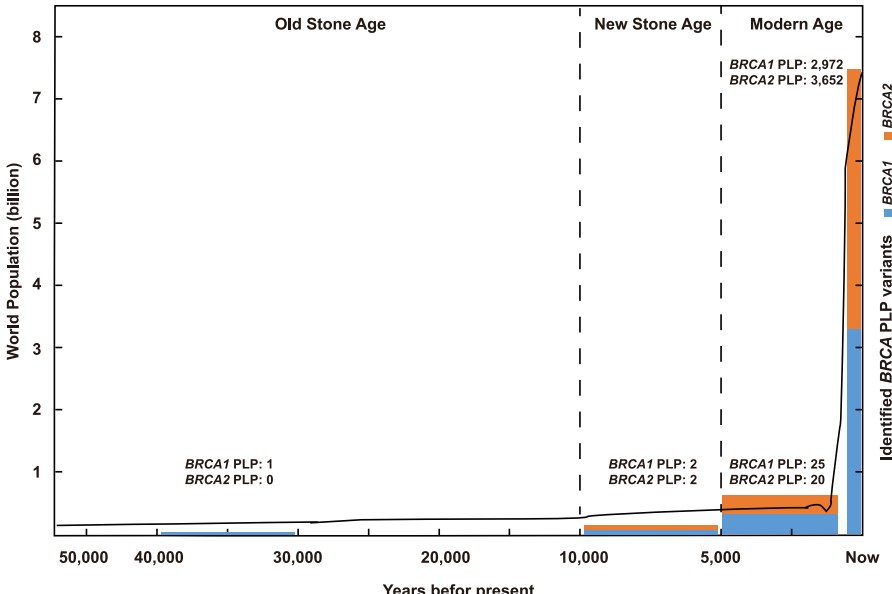

**Figure 4. Model for the origin of human *BRCA* PLP.**
It shows that the positive selection on human *BRCA* causes high human *BRCA* variation and accompanied PLP variation, and the expansion of human population further increases the spectrum of human *BRCA* PLP. X-axis: the timing of modern human after out-of-Africa migration; Left y-axis: size of human population; Right y-axis: number of human *BRCA* PLP variants. Curve: the growth of human population in the past 50,000 yr.

mainly cause high cancer risk after reproductive age. Homozygotic *BRCA* PLP carriers seldom survive or develop Fanconi anemia due to the embryonic lethal effect, as BRCA is essential for development (Seo et al, 2018).

A limitation of our study is that most of the *BRCA* founder PLP variants and ethnic distribution data used in the study were derived from European and descendants. More data from non-European populations should enhance the significance of our study.

## Materials and Methods

### Source of human *BRCA* PLPs

Human *BRCA* PLP variants were from ClinVar database (https://ftp.ncbi.nlm.nih.gov/pub/clinvar/vcf_GRCh38/weekly/clinvar_20201031.vcf.gz, accessed 1 November 2020). Only the variants classified as "pathogenic," "likely pathogenic," and "pathogenic and likely pathogenic" were included in the study. The variants with conflict classifications were excluded to ensure the reliability of the analysis.

### Vertebrate genomic mapping analysis

Human *BRCA* PLP variants were divided into single nucleotide variants and indel groups. Their genomic positions were annotated by referring to the following reference sequences: *BRCA1*: genome hg38 NC_000017.11, cDNA NM_007294, NP_009225; *BRCA2*: genome hg38 NC_000013.11, cDNA NM_000059, NP_000050. *BRCA1*, and *BRCA 2* sequences across 100 vertebrate species were downloaded from UCSC Genome Browser (https://genome.ucsc.edu/cgi-bin/hgGateway?redirect=manual&source=genome.ucsc.edu). Sequence alignment followed the procedures in Multiz Alignments of 100 Vertebrates part in UCSC genome browser (https://genome.ucsc.edu/cgi-bin/hgc?db=hg38&c=chr17&l=43106526&r=43106527&o=43106526&t=43106527&g=multiz100way&i=multiz100way). Tree model of the 100 vertebrate species was generated using the phyloFit program in the PHAST package (Murphy et al, 2001). The phylogenetic tree for the 100 vertebrate species was from UCSC resource (http://hgdownload.soe.ucsc.edu/goldenPath/hg38/multiz100way/), and the distance between species on the tree was adjusted to ensure the readability in the figures. PhastCons and phyloP from the PHAST package were used for evolution conversion measurement. Sequence alignment between repeat-masked human hg38 and non-human genome sequences was made by using Lastz (BLASTZ) and multiz (Blanchette et al, 2004; Hubisz et al, 2011; Armstrong et al, 2019; Ramani et al, 2019). Based on the phylogenetic distance from the references, the scoring matrix and parameters for pairwise were adjusted for each species. The high-score chains were placed along genome sequences, and the low-score chains were used to fill gaps. The results of single base-level alignment were collected from the "Multiz Alignments of 100 Vertebrates" section in UCSC genome browser by entering variant position in hg38 using a Python-based tool (https://github.com/Skylette14/GetBase). For indel alignments, the base number of insertion or deletion between human and other species was the same. Each indel alignment across the matched species were manually checked to ensure reliability of the alignment. The positions corresponding to human PLP variants in the aligned non-human sequences were obtained by using GetBase.

### Ancient human genomic mapping analysis

Ancient human DNA sequences and related publications were from "Allen Ancient DNA Resource (version 42.2, https://reich.hms.harvard.edu/allen-ancient-dna-resource-aadr-downloadable-genotypes-present-day-and-ancient-dna-data, accessed 1 March 2020), containing genomic sequences from a total of 2,792 ancient human individuals dated from 37,470 to 300 before present.

Bam files of ancient genomic sequences were downloaded. The sequences containing *BRCA1* (chr17:41,196,312-41,277,500, hg19 by Ensembl) and *BRCA2* (chr13:32,889,611-32,973,805, hg19 by Ensembl) were identified and used for mapping to hg19. Mpileup command in SAMtools was used for variant calling from the mapped sequences with the minimal base quality set as 1 (Li et al, 2009). After generation of ancient human vcf files, the called variants were annotated using wANNOVAR (https://wannovar.wglab.org/) (Chang & Wang, 2012), compared with human *BRCA* PLP variants and manually checked in ClinVar to obtain the related information for these matched by the ancient *BRCA* sequences. The locations for the ancient PLP carrier's fossil excavation and the estimated age were based on the original publications. The geographical distribution map of the ancient *BRCA* PLP variants were generated by using Matlab (The MathWorks, Inc.).

## Statistical analysis

Kruskal–Wallis test was used for statistical comparison for the *BRCA* PLP variants shared between clades (GraphPad Software). Chi-Squared test was used to compare the distribution of the shared *BRCA* PLP variants, $P < 0.01$ was considered as statistical significance.

# Data Availability

All data used in the study were from public domains as indicated in the text. The data generated from the study were provided as online Figs S1–S3 and Tables S1–S6.

# Supplementary Information

# Acknowledgements

We thank Information and Communication Technology Office (ICTO), University of Macau for providing the High-Performance Computing Cluster resources for the study. This work was supported by the grants from Macau Science and Technology Development Fund (085/2017/A2, 0077/2019/AMJ), the grants from the University of Macau (SRG2017-00097-FHS, MYRG2019-00018-FHS), the Faculty of Health Sciences, University of Macau (Startup fund, FHSIG/SW/0007/2020P, FHS Innovation grant) (SM Wang).

## Author Contributions

J Li: data curation, software, formal analysis, visualization, methodology, and writing—original draft, review, and editing.
B Zhao: data curation, software, methodology, and writing—original draft, review, and editing.
T Huang: software and writing—original draft, review, and editing.
Z Qin: data curation, software, and writing—original draft, review, and editing.
SM Wang: conceptualization, resources, funding acquisition, project administration, and writing—original draft, review, and editing.

## Conflict of Interest Statement

The authors declare that they have no conflict of interest.

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
