## [Reviewer comments · Life Science Alliance]

Life Science Alliance

Human BRCA pathogenic variants were originated during recent human history

Jiaheng Li, Bojin Zhao, Teng Huang, Zixin Qin, and San Ming Wang

DOI: <https://doi.org/10.26508/lsa.202101263>

Corresponding author(s): San Ming Wang, University of Macau

Review Timeline:

Submission Date:	2021-10-15
Editorial Decision:	2021-12-06
Revision Received:	2021-12-10
Editorial Decision:	2022-01-17
Revision Received:	2022-01-21
Editorial Decision:	2022-01-21
Revision Received:	2022-01-25
Accepted:	2022-01-26

Scientific Editor: Novella Guidi

Transaction Report:

December 6, 2021

Re: Life Science Alliance manuscript #LSA-2021-01263-T

Prof. San Ming Wang
Cancer Centre and Institute of Translational Medicine, Faculty of Health Sciences, University of Macau
University Avenue
Taipa 999078
Macau

Dear Dr. Wang,

Thank you for submitting your manuscript entitled "Human BRCA pathogenic variants were originated during recent human history" to Life Science Alliance. The manuscript was assessed by expert reviewers, whose comments are appended to this letter. As you will note from the reviewers' comments below, the reviewers are overall interested in this study but they do raise important concerns that need to be addressed before resubmission with the main ones being the many errors in syntax, spelling and phrasing, and the poor description of the methods and results that make it difficult to verify the conclusions. We, thus, encourage you to submit a revised version of the manuscript back to LSA that responds to all of the reviewers' points.

Thank you for this interesting contribution to Life Science Alliance. We are looking forward to receiving your revised manuscript.

Sincerely,

B. MANUSCRIPT ORGANIZATION AND FORMATTING:

Reviewer #1 (Comments to the Authors (Required)):

This brief analysis of BRCA1 and BRCA2 mutation patterns across human pre-history is an interesting analysis of the patterns of emergence of pathogenic mutations of BRCA genes. As noted by the authors, the relatively high frequency of pathogenic or likely pathogenic (PLP) BRCA alleles is not well understood. The authors summarize two possible explanations: "Possible explanations can be 1). BRCA PLPs develop cancer mostly at later reproduction [sic] or menopause [sic] stage. Before reaching the stage, the mutation has already transmitted to the next generation; 2). The oncogenic effects of BRCA PLPs could be developmental stage-dependent. They could be beneficial [sic] at the reproductive stage but deleterious at later reproduction [sic] or menopause [sic] stage. As positive selection mostly imposes on reproductive stage, BRCA PLPs may be favorably [sic] selected." The analysis presented does not resolve this question, yet in the body of the text it is suggested that positive selection for BRCA PLP alleles did indeed occur. It is important to clarify the logic and to stick to one point of view-the correct one being that the issue is unresolved.

As is clear from the excerpt quoted above, the phrasing, spelling and syntax contains numerous errors, making the arguments sometimes difficult to follow. It would be worthwhile to enlist help in rewriting the m/s to make the text clearer and to correct the numerous errors. In other respects, this reviewer found the summary of BRCA gene PLP mutations across human (pre-)history interesting and informative, as was the comparison across species.

Reviewer #2 (Comments to the Authors (Required)):

The manuscript describes a highly interesting analysis of pathogenic and likely pathogenic variants in the BRCA1/2 genes across species and in ancient human DNA. The very interesting conclusion is that the mutations are in general young - less than 3500 years.

The comprehensive cross specie and ancient analysis is original and interesting.

However, there are some major questions to the paper:

- 1) the authors describe positive selection for PLP variants. This is controversial and should be explained and justified in more detail. Obviously, there is a negative selection against PLPs since dual carriers do not survive or develop Fanconi Anemia. This should be discussed.
- 2) a more likely explanation for the many different PLPs is that the genes are large. The reason why the mutations are young could be the negative selection. This should also be discussed.
- 3) why are there many mutations common to other species especially avians, despite not being conserved in evolution? The most likely explanation is mutational hotspots. This may not be expected or loss of function mutations.
- 4) page 9. Results from phylogenetic analysis are referred to but not shown. It may be obvious, but it would be illustrative to show.
- 5) page 14: the oncogenes effect of BRCA PLP. BRCA genes are tumor suppressors, not oncogenes.
- 6) figure 3. Yellow color is hardly visible.

Reviewer #3 (Comments to the Authors (Required)):

In this manuscript Li and colleagues examine the origins of BRCA pathogenic/likely pathogenic variants in human history. The paper is essentially divided in two parts. The first of which uses phylogenetic analysis to determine the origins of pathogenic variants; whether they may have originated in common evolutionary ancestors and its presence in current human populations would have been driven by positive selection. The second searches for pathogenic/likely pathogenic variants in ancient human samples of a wide range of dates (dated from 300 to 45,000 years ago).

The manuscript has several issues that need to be addressed.

Overall, the methods, description of results, and underlining data are so brief that it is extremely difficult to understand the

rationale behind the approach, what was done with the data and to verify that the conclusions are warranted by the results obtained.

For example, the authors state that they 'searched' for human PLP in the reference (presumably) sequence of other species, but it is unclear what they mean by that. It is not described how the sequences were aligned and the criteria for the inclusion of species, or whether it was disproportionately skewed to mammalian species. There is no information about how the alignments were inspected whether there was manual editing, whether the alignment was also done at the level of amino acid. The authors identify a significant fraction of species that carry variants corresponding to human pathogenic variants. Because these are overwhelmingly premature termination variants and most of them lead to inactive proteins, are we to assume that many species do not have functional BRCA? I couldn't find the alignments (or even representative examples of the alignments) to examine the results. The excel table with a selection of nucleotides that are proposed to represent these variants lacks context and details (accession number, etc).

The few specific examples are problematic: the three pathogenic variants in BRCA1 presented as intronic (pages 7 and 8) are not intronic but rather coding substitutions, and the BRCA2 variant presented as pathogenic shared with 57 species (p.Met1168Ile) is not pathogenic, but rather a VUS. Conclusions are presented in a very confusing way as exemplified by the self-contradictory statements: "We found no evidence for their presence in these species, but the wildtype alleles in BRCA1 185delAG was present across 51 species from Chimpanzee to Armadillo, in BRCA1 5382insC across 83 species from Rhesus to Coelacanth, in BRCA2 6174delT across 87 species from Chimpanzee to Tetraodon." Therefore, the results presented in this section seems unconvincing at best.

The section on ancient DNAs suffers from the same issues and alignments are not provided. It is unclear whether the sequences in which they were identified are part of complete sequence of the locus or from smaller fragments. Here the authors find a very high frequency of individuals with pathogenic variants (1.8%) including findings of variants rare in current populations being found in more than one individual in a subpopulation. One possibility is that this is an extremely small sample to derive conclusions. In this context, statements such as "The PLPs shared between the much early Eurasians and the much later Dominicans implies that these could be inherited from Europeans to Dominicans in recent colonization history." seem unwarranted.

It is unclear how they determined the 'time of origin' as they state that they conducted "haplotyping analysis", but very little information is given. No consideration is given to the possibility that mutations may arise at multiple times in multiple independent populations. Also, no consideration is given to the role of pleiotropy in driving selection of variants whose effects mostly occur in post-reproductive age.

Minor issues

The section on "Data from previous studies" seems dispensable and can be summarized in the introduction.

Please define the criteria ('interpretation'; 'review status'; how to deal with variants that have multiple interpretation, etc) used to extract the data from ClinVar. This is extremely important as it is the basis of the analysis. State the date and version (if applicable) when the data was extracted.

Please define what is being considered 'Pathogenic' and 'Likely Pathogenic' (ClinVar? ENIGMA consortium data?).

Point-to-point response to reviewers' comments**Reviewer #1****Question**

As noted by the authors, the relatively high frequency of pathogenic or likely pathogenic (PLP) BRCA alleles is not well understood. The authors summarize two possible explanations:

"Possible explanations can be 1). BRCA PLPs develop cancer mostly at later reproduction or menopause stage. Before reaching the stage, the mutation has already transmitted to the next generation; 2). The oncogenic effects of BRCA PLPs could be developmental stage-dependent. They could be beneficial at the reproductive stage but deleterious at later reproduction or menopause stage. As positive selection mostly imposes on reproductive stage, BRCA PLPs may be favorably selected."

The analysis presented does not resolve this question, yet in the body of the text it is suggested that positive selection for BRCA PLP alleles did indeed occur. It is important to clarify the logic and to stick to one point of view-the correct one being that the issue is unresolved.

Answer

Thanks for the comments. Indeed, our study doesn't provide answer for the two possible explanations.

In the revision, the sentences were included in Discussion to clarify the issue:

Possible explanations can be 1). The cancer caused by *BRCA* PLPs occurs mostly at later reproduction stage. Before reaching the stage, the mutation has already been transmitted to the next generation; 2). The loss of tumor suppressing function of *BRCA* due to *BRCA* PLPs could be developmental stage-dependent that the PLPs could be beneficial at the reproductive stage but deleterious at later reproduction stage. Positive selection imposed on reproductive stage can select the *BRCA* PLPs not deleterious at the stage. However, our current study does not determine whether one of the two explanations or both could contribute to the higher frequency of *BRCA* PLP in human population.

Question

As is clear from the excerpt quoted above, the phrasing, spelling and syntax contains numerous errors, making the arguments sometimes difficult to follow. It would be worthwhile to enlist help in rewriting the m/s to make the text clearer and to correct the numerous errors.

Answer

Thanks for the comments. In the revision, we have made extensive correction and a native English speaker colleague has polished the final manuscript. We hope the resubmitted version has improved the quality of English writing.

Reviewer #2**Question**

1) the authors describe positive selection for PLP variants. This is controversial and should be explained and justified in more detail. Obviously, there is a negative selection against PLPs since dual carriers do not survive or develop Fanconi Anemia. This should be discussed.

Answer

Thanks for the comments. Rich data from BRCA evolution study in the past 2 decades has well demonstrated the presence of two types of evolution selection in BRCA: the positive selection in human, chimp and bonobo, and the negative/neutral selections in all other species. As described in our manuscript “Studies revealed that *BRCA* is evolutionarily highly conserved across wide-range species, including primates, by negative selection in reflecting its essential roles in maintaining genome stability; it has also observed that *BRCA* is rapidly evolving by positive selection in humans and its close living relatives of the Chimpanzee and Bonobo (Huttley et al. 2000; Fleming et al. 2003; Abkevich et al. 2004; Pavlicek et al. 2004; Burk-Herrick et al. 2006; O’Connell 2010; Lou et al. 2014). Besides our current study in human BRCA, we also did a parallel evolution study in mouse *Brca*. Our study showed that negative/neutral selection is present in mouse *Baca* (Wang and Wang 2021). The positive selection likely reflects the new function of *BRCA* gained in these three species of human, chimp and bonobo (Huttley et al. 2000)”. It is true that “dual carriers do not survive or develop Fanconi Anemia”. This is more likely due to the embryonic lethal effects of homozygotic BRCA mutation as BRCA is essential for proper development (Xu, X. et al. Nat Genet 22, 37-43, 1999).

In the revision, the paragraph has been revised as the following paragraph to clarify better the issue:

Studies revealed that *BRCA* across wide-range species including most of the primates is evolutionarily highly conserved by negative selection in reflecting its essential roles in maintaining genome stability. In contrast, however, *BRCA* in humans and its closest living relatives of the Chimpanzees and Bonobos is rapidly evolving by positive selection (Huttley et al. 2000; Fleming et al. 2003; Abkevich et al. 2004; Pavlicek et al. 2004; Burk-Herrick et al. 2006; O’Connell 2010; Lou et al. 2014, Wang et al, 2021). The positive selection likely reflects the new function of *BRCA* gained in these three species, such as promoting immunity to counter viral infection (Lou et al. 2014), regulating gene expression and metabolism (Rosen et al. 2006; Chen et al. 2020), enhancing neural development (Pao et al. 2014), and increasing reproduction (Smith et al. 2013).

The following paragraph is included under Discussion:

Homozygotic BRCA PLP carriers seldom survive or develop Fanconi Anemia (Seo et al. 2018). This is due to the embryonic lethal effect of homozygotic *BRCA* mutation as BRCA is essential for development rather than negative selection against *BRCA* PLPs.

Question

2) a more likely explanation for the many different PLPs is that the genes are large. The reason why the mutations are young could be the negative selection. This should also be discussed.

Answer

Thanks for the comments. Based on current knowledge, human BRCA is under positive selection but not negative selection (Huttley et al. 2000; Fleming et al. 2003; Abkevich et al. 2004; Pavlicek et al. 2004; Burk-Herrick et al. 2006; O'Connell 2010; Lou et al. 2014). This is well demonstrated by the Figure 2 in the pioneer BRCA phylogenetic study, showing the positive selection in human and chimpanzee BRCA1, and negative selection in other species (Huttley GA et al. Nat Genet 25: 410-413, 2000). The observation has been confirmed by many following reports including ourselves.

Human BRCA1 has over 1,863 residues, and human BRCA2 has over 3,418 residues. While the large size of human BRCA can be a factor, it is very unlikely that it can be a major factor contributing to the large quantity of PLPs. As shown by our own study, mouse Brca is highly stable without much variation although the size of mouse Brca (Brca1 1,812 residues, Brca2 3,329 residues) are very close to the human ones.

In the revision, the following paragraph has been included in the Discussion to explain this issue:

Other factors may also contribute to the rich human *BRCA* PLPs. Human *BRCA1* has 1,863 residues and *BRCA2* has 3,418 residues. Although gene with large size can produce more genetic variants, it is unlikely that the size in *BRCA* can be a major factor for the large quantity of BRCA PLPs. Our study showed that mouse Brca does not contain much variation although the size of mouse Brca (Brca1 1,812 residues, Brca2 3,329 residues) is very close to the human BRCA (Wang et al, 2021).

Question

3) why are there many mutations common to other species especially avians, despite not being conserved in evolution? The most likely explanation is mutational hotspots. This may not be expected or loss of function mutations.

Answer

We do not have explanation for the “far distance BRCA PLP conservation” we named between human and Aves and Sarcopterygii. A study (Gao L and Zhang J. Why are some human disease-associated mutations fixed in mice? TRENDS in Genetics. 19.12: 678-681, 2003)

compared multiple human deleterious mutations, including several mutations in *BRCA*, shared with mouse. They evaluated multiple theories including “Founder effect”, “Fixations of slightly deleterious mutations”, “Relaxed selection on late-onset phenotypes” and “Compensatory changes”, they were in favor of the “compensation theory”, which states that “compensatory mutations at other sites of the same or a different protein render the deleterious mutations neutral”, to explain the conservation of human deleterious mutation across distant species. We consider that the compensation theory can also be used to explain the sharing of human *BRCA* PLPs in aves, although data is not available to validate the explanation.

In the revision, the entire paragraph has been modified to clarify the issue:

Many human *BRCA* PLPs were shared with species in Aves and Sarcopterygii clads. This could happen by chance rather than by cross-species evolution conservation, as it is unlikely that the conservation would allow the presence of the gaps across such wide distances. Alternatively, it could also be related with species-specific pathogenicity of the same *BRCA* variants that the *BRCA* PLPs in humans may not be pathogenic in non-human species (Gao and Zhang, 2003). There is lack of evidence to link the cancers in non-human species with their shared human *BRCA* PLPs. For example, Tasmanian devil has high risk to develop facial cancer but there is no evidence to link the disease with the multiple human *BRCA1* PLPs its shared (Hamilton 1966). A study analyzed multiple human deleterious mutations, including several mutations in *BRCA*, shared with mouse (Gao and Zhang. 2003). Among multiple theories including “Founder effect”, “Fixations of slightly deleterious mutations”, “Relaxed selection on late-onset phenotypes” and “Compensatory changes”, they considered that the “compensation changes”, which stated that “compensatory mutations at other sites of the same or a different protein render the deleterious mutations neutral”, can be the best to explain the conservation of human deleterious mutation with the distant species of mouse. We consider that the compensation theory can also be used to explain the sharing of human *BRCA* PLPs with the species in aves and Sarcopterygii clads, although direct evidence is not available to validate the explanation.

Question

4) page 9. Results from phylogenetic analysis are referred to but not shown. It may be obvious, but it would be illustrative to show.

Answer

Sorry for the confusion. The sentence “The results from the phylogenetic analysis didn’t support evolution conservation as the main source of human *BRCA* PLPs.” is to make conclusion for the phylogenetic study.

In the revision, the sentence has been revised as:

The results from our phylogenetic analysis above didn’t support evolution conservation as the source of human *BRCA* PLPs.

Question

5) page 14: the oncogenes effect of BRCA PLP. BRCA genes are tumor suppressors, not oncogenes.

Answer

Sorry for the confusion. The original sentence “The oncogenic effects of *BRCA* PLPs could be developmental stage-dependent.” refers to the oncogenic effected by the mutated BRCA but not the wildtype BRCA.

In the revision, the sentence has been changed as:

The loss of tumor suppressing function of BRCA due to BRCA PLPs could be developmental stage-dependent.

Question

6) figure 3. Yellow color is hardly visible.

Answer

To make better visible for the figure, the color range used in the figure has been changed from yellow-to-blue to red-to- blue. In this way, the original yellow spot is changed to red spot.

Reviewer #3

Question

Overall, the methods, description of results, and underlining data are so brief that it is extremely difficult to understand the rationale behind the approach, what was done with the data and to verify that the conclusions are warranted by the results obtained. For example, the authors state that they 'searched' for human PLP in the reference (presumably) sequence of other species, but it is unclear what they mean by that. It is not described how the sequences were aligned and the criteria for the inclusion of species, or whether it was disproportionately skewed to mammalian species. There is no information about how the alignments were inspected whether there was manual editing, whether the alignment was also done at the level of amino acid.

Answer

Thanks for the comments. Indeed, the methodological description in the original manuscript was rather brief, as we thought that many methodologies we used followed the well-established approaches by others, such as phylogenic analysis and archeological analyses. Obviously, too briefs can cause difficulties for readers to follow up the details as indicated by the comments made by the reviewer.

In the revision, we expanded the description including more technique details and adding citations/links to help readers to better understand the rationale and conditions used in the analysis.

Question

The authors identify a significant fraction of species that carry variants corresponding to human pathogenic variants. Because these are overwhelmingly premature termination variants and most of them lead to inactive proteins, are we to assume that many species do not have

functional BRCA? I couldn't find the alignments (or even representative examples of the alignments) to examine the results. The excel table with a selection of nucleotides that are proposed to represent these variants lacks context and details (accession number, etc).

Answer

Thanks for the comments. In the revision, we have added the actual human BRCA PLP variants and ClinVar allele ID at the top of Supplemental Table 3 (BRCA1) and Supplemental Table 4 (BRCA2) to allow direct comparison between species; we have also generated Supplementary Figure 1 showing examples of the alignments.

Question

The few specific examples are problematic: the three pathogenic variants in BRCA1 presented as intronic (pages 7 and 8) are not intronic but rather coding substitutions, the BRCA2 variant presented as pathogenic shared with 57 species (p.Met1168Ile) is not pathogenic, but rather a VUS.

Answer

Very sorry for the mistake: "Intronic" is a typo.

In the revision, the sentence has been revised as:

"For example, *BRCA1* c.3268C>T (p.Gln1090Ter) was shared with 19 species from Rock pigeon in Aves to Lizard in Sarcopterygii, c.2498T>A (p.Leu833Ter) with 18 species from Saker falcon in Aves to Spiny softshell turtle in Sarcopterygii, and c.2138C>A (p.Ser713Ter) with 18 species from Rock pigeon in Aves to Lizard in Sarcopterygii".

The ClinVar BRCA PLPs used in our study was accessed on November1, 2020. BRCA2 c.3504G>T (p.Met1168Ile) was then classified as "Likely_pathogenic". In the new ClinVar classification (access on December 8, 2021), BRCA2 c.3504G>T (p.Met1168Ile) has been re-classified as VUS.

In the revision, we used the pathogenic *BRCA2* c.8933C>A (p.Ser2978Ter) to replace *BRCA2* c.3504G>T (p.Met1168Ile) to avoid this trouble, and the sentence is revised as:

c.8933C>A (p.Ser2978Ter) with 36 species from Lesser Egyptian jerboa in Euarchontoglires to Opossum in Mammal, and the most diatal species shared with this variant was Spotted gar in Fish.

Question

Conclusions are presented in a very confusing way as exemplified by the self-contradictory statements: "We found no evidence for their presence in these species, but the wildtype alleles in BRCA1 185delAG was present across 51 species from Chimpanzee to Armadillo, in BRCA1 5382insC across 83 species from Rhesus to Coelacanth, in BRCA2 6174delT across 87 species from Chimpanzee to Tetraodon." Therefore, the results presented in this section seems unconvincing at best.

Answer

Very sorry for the confusion here. What we want to say is that the wildtype bases in the three typical BRCA PLP are highly conserved without variation.

In the revision, the paragraph has been revised as the followings:

As control, we searched *BRCA1* c. 68_69del, *BRCA1* c.5266dup and *BRCA2* c.5946del, the three *BRCA* founder PLPs in Ashkenazi Jews population, in the 100 vertebrates (Levy-Lahad et al. 1997). We found no evidence for their presence in these species, but the wildtype AG at the position of 68-69 in *BRCA1* was present across 51 species from Chimpanzee to Armadillo, the wildtype C at the position of 5266 in *BRCA1* was present across 83 species from Rhesus to Coelacanth, the wildtype T at the position of 5946 of *BRCA2* was present across 87 species from Chimpanzee to Tetraodon.

Question

The section on ancient DNAs suffers from the same issues and alignments are not provided. It is unclear whether the sequences in which they were identified are part of complete sequence of the locus or from smaller fragments. Here the authors find a very high frequency of individuals with pathogenic variants (1.8%) including findings of variants rare in current populations being found in more than one individual in a subpopulation. One possibility is that this is an extremely small sample to derive conclusions. In this context, statements such as "The PLPs shared between the much early Eurasians and the much later Dominicans implies that these could be inherited from Europeans to Dominicans in recent colonization history." seem unwarranted.

Answer

Following the comments, we generated Supplementary Figure 1A to include multiple alignment examples between modern and ancient *BRCA* PLP. Indeed, the number of ancient cases being sequenced is much smaller than the number of modern cases being sequenced. This is exemplified by the availability of only 3 Neanderthal and 2 Denisovan genomes over 40,000 year-old.

"It is unclear whether the sequences in which they were identified are part of complete sequence of the locus or from smaller fragments.": As shown in Supplementary Table 2, of the 2,972 *BRCA1* PLP, less than 10 were indels with >50base change, and most were single base variation or smaller indel with a few bases; similar situation was present in 3,652 *BRCA2* PLP. Therefore, the sequences in which they were identified are mostly single base changes.

"The very high frequency of individuals with pathogenic variants (1.8%) including findings of variants rare in current populations being found in more than one individual in a subpopulation": Indeed, the match is much likely by chance rather than statistically significant events.

In the revision, the sentence "The PLPs shared between the much early Eurasians and the much later Dominicans implies that these could be inherited from Europeans to Dominicans in recent colonization history." has been revised to let the observation less conclusive:

The PLP shared between the much early Eurasians and the much later Dominicans suggests that these might be inherited from Europeans to Dominicans in recent colonization history.

Question

It is unclear how they determined the 'time of origin' as they state that they conducted "haplotyping analysis", but very little information is given.

Answer

Thanks for the comments that we should have made better explanation on the issue. The haplotyping assay is widely used in *BRCA* founder mutation identification with their age determined, for example, the age of the three *BRCA* founder mutations (*BRCA1* c. 68_69del and c.5266dup and *BRCA2* c.5946del in Ashkenazi Jews population. Haplotyping assay ensures that the variant is inherited by referring the targeted variant with other neighboring variants together as a block. In our study, we directly used the haplotyping assay-determined *BRCA* founder mutations from literature but we didn't perform haplotyping testing ourselves. The principle of using haplotyping to estimate age of genetic variants was developed decades ago. Here, we cite the description in a paper (PMID: 17902052) outlining how it works: "this method estimates the number of generations between the present time and the most recent common ancestor of all the mutation-bearing individuals as a function of the observed haplotypes, the recombination fraction between the mutation and the marker loci (and markers themselves), the rate of mutation at the microsatellite and SNP marker loci, and the allele frequencies of the marker loci. The 95% confidence interval for estimated allele age was calculated from the maximum likelihood estimate.". Our study observed that the known human *BRCA* founder mutations were arisen 3,225 to 140 years ago.

In the revision, the following sentences are included in "Dated *BRCA* founder PLPs and ethnic-specific distribution of *BRCA* PLPs" under Results:

Many *BRCA* PLPs have been determined as *BRCA* founder variants in different ethnic population, their arisen times were determined by haplotyping analysis. For example, the three *BRCA* founder mutations (*BRCA1* c. 68_69del and c.5266dup and *BRCA2* c.5946del in Ashkenazi Jews population were determined being arisen 1,720, 1,800, and 580 years ago respectively (Table 3).

Question

No consideration is given to the possibility that mutations may arise at multiple times in multiple independent populations.

Answer

Thanks for the valuable comments. It is certainly possible that some *BRCA* PLP could be arisen by chance at the same genetic locations in different populations.

In the revision, the following sentences are added to indicate the possibility under Discussion:

Another factor related with ethnic-specific PLPs may be that the same *BRCA* PLP could arose by chance in different populations at different times. For example, the *BRCA1* 68-69del founder mutation in Ashkenazi Jews is also present in other populations at very low frequencies, and *BRCA1* 3607C>T shared between the Slovenia case dated 4971±450 ago and Dominica case

dated 921±500 ago (Table 2). However, this unlikely played a major role in contributing to human BRCA PLPs.

Question

Also, no consideration is given to the role of pleiotropy in driving selection of variants whose effects mostly occur in post-reproductive age.

Answer

This is certainly possible. George William's pleiotropy theory indeed provide another anchor to look at the origin of evolution selection for human BRCA PLPs, particularly for these selected in post-reproductive age.

In the revision, we added the following sentences to reflect this factor under **Discussion**:

In addition, pleiotropy effects may also exist that variation in none-BRCA genes may contribute to the selection of BRCA PLPs, particularly for these selected in the post-reproductive age (Williams. 1957). This possibility is interesting as BRCA PLP mainly cause high cancer risk after reproduction age.

Question

The section on "Data from previous studies" seems dispensable and can be summarized in the introduction.

Answer

Fully agree. In the revision, the entire section is relocated to the introduction part.

Question

Please define the criteria ('interpretation'; 'review status'; how to deal with variants that have multiple interpretation, etc) used to extract the data from ClinVar. This is extremely important as it is the basis of the analysis. State the date and version (if applicable) when the data was extracted.

Answer

This is certainly an important issue for the study. The *BRCA* PLP data used in our study (Supplementary Table 2) was originated from ClinVar. We only used the *BRCA* PLP with single classification as P or LP. These with conflict interpretation, multiple interpretation, and uncertain classification were all excluded from our study to ensure the reliability of the analysis.

In the revision, the following sentence is added to clarify the issue:

The variants with conflict classifications were excluded from our study to ensure the reliability of the analysis.

Question

Please define what is being considered 'Pathogenic' and 'Likely Pathogenic' (ClinVar? ENIGMA consortium data?).

Answer

ClinVar classification of 'Pathogenic' and 'Likely Pathogenic' is based on a consortium activity, involving the evidence from ENIGMA, expert panel, ACMG guidelines, Clinical testing labs, etc. ENIGMA is one of the major sources but not the only one.

January 17, 2022

Re: Life Science Alliance manuscript #LSA-2021-01263-TR

Prof. San Ming Wang
University of Macau
Faculty of Health Sciences
University of Macau
Taipa
Macau SAR, Taipa 999078
Macao

Dear Dr. Wang,

Thank you for submitting your revised manuscript entitled "Human BRCA pathogenic variants were originated during recent human history" to Life Science Alliance. The manuscript has been seen by two of the original reviewers whose comments are appended below. While the reviewers continue to be overall positive about the work in terms of its suitability for Life Science Alliance, some important issues remain. As you will note from the reviewers' comments below, Reviewer 3 still has some remaining concerns that need to be addressed before resubmission. In the light of this, we encourage you to make a more careful update of the references and argument better why you think there is a high prevalence of PLPs. Also toning down the conclusions would be beneficial to avoid being misleading.

Our general policy is that papers are considered through only one revision cycle; however, given that the suggested changes are relatively minor, we are open to one additional short round of revision. Please note that I will expect to make a final decision without additional reviewer input upon resubmission.

Please submit the final revision within one month, along with a letter that includes a point by point response to the remaining reviewer comments.

To upload the revised version of your manuscript, please log in to your account: <https://lsa.msubmit.net/cgi-bin/main.plex>
You will be guided to complete the submission of your revised manuscript and to fill in all necessary information.

- A letter addressing the reviewers' comments point by point.
- An editable version of the final text (.DOC or .DOCX) is needed for copyediting (no PDFs).
- High-resolution figure, supplementary figure and video files uploaded as individual files: See our detailed guidelines for preparing your production-ready images, <https://www.life-science-alliance.org/authors>
- Summary blurb (enter in submission system): A short text summarizing in a single sentence the study (max. 200 characters including spaces). This text is used in conjunction with the titles of papers, hence should be informative and complementary to the title and running title. It should describe the context and significance of the findings for a general readership; it should be written in the present tense and refer to the work in the third person. Author names should not be mentioned.

B. MANUSCRIPT ORGANIZATION AND FORMATTING:

Sincerely,

Reviewer #2 (Comments to the Authors (Required)):

All queries have been answered satisfactorily. The paper brings interesting new knowledge to the field.

Reviewer #3 (Comments to the Authors (Required)):

While the paper was improved in adding additional information about the methods. However, many of the reviewers questions were not satisfactorily answered.

In particular, the question about whether these species that carry inactivating mutations have no functional BRCA was not answered.

Many citations are not directly related to the statements.

The treatment of variants that are considered pathogenic is not reassuring. The premise that the frequency of BRCA PLPs are high is, in my view, incorrect, and no evidence is shown to support the numbers. In fact, only in high risk cohorts the prevalence approaches the numbers cited (and implicitly stated to correspond to unselected cohorts).

The approach to compare large population datasets with sequence of homologs as presented seems to lack rigor and the conclusions are not compelling. The authors continue to make broad-stroke conclusions with references that are clearly inappropriate. The passage referring to the Tasmanian Devil transmissible face tumor (which has been well characterized and is a unrelated tumor) with a reference from thirty years before the cloning of BRCA1, is just one example.

Reply to editor's comments

Reviewer 3 still has some remaining concerns that need to be addressed before resubmission. In the light of this, we encourage you to

1. make a more careful update of the references

Answer

We have double-checked all references, replaced the less relevant ones with more relevant ones, and added new ones to the topics as marked in the revision.

2. argument better why you think there is a high prevalence of PLPs.

Answer

We have increased the scope in Discussion to further address the issue.

3. Also toning down the conclusions would be beneficial to avoid being misleading.

Answer

Thanks for indicting the issue. We have modified the wording for the conclusion to avoid overstate the data.

Reply to reviewer #3 comments**Question**

whether these species that carry inactivating mutations have no functional BRCA was not answered.

Answer

There are rich evidence showing the consequences caused by the inactivating mutations in non-human *BRCA*, as well reflected by the studies in *Brca1* knockout mouse studies performed in the past 3 decades by different laboratories around the world. The conventional *Brca1* knockout in mice caused post-implantation development (1), embryonic cellular proliferation (2), embryonic lethal (3), neuroepithelial abnormalities (4); and the conditional *Brca1* knockout in mice led to gamma-irradiation hypersensitivity and genetic instability (5), abnormal T cell development (6, 7), and tumorigenesis (8, 9).

In the revision, the following sentences and references are included to address the issue:

Extensive animal model studies have provided rich evidence in showing the pathogenic consequences by *BRCA* PLP in non-human species, especially from mouse studies. *Brca1* knockout-mice showed abnormal post-implantation development and embryonic proliferation (Liu et al. 1996, Hakem et al. 1996), embryonic lethality (Ludwig et al. 1997), neuroepithelial abnormalities (Gowen et al. 1996), irradiation hypersensitivity and genetic instability (Shen et al. 1998), abnormal T cell development (Mak et al. 2000; Xu et al. 2001), and tumorigenesis (Xu et al. 1999; Brodie et al. 2001). Without the presence of other mutated genes such as the mutated TP53, however, the mutated *Brca1* alone is not sufficient to directly cause cancer (Xu et al. 2001).

Related references

Liu CY, Flesken-Nikitin A, Li S, Zeng Y, Lee WH. Inactivation of the mouse *Brca1* gene leads to failure in the morphogenesis of the egg cylinder in early postimplantation development. *Genes Dev* 1996; 10:1835-43.

Hakem R, de la Pompa JL, Sirard C, Mo R, Woo M, Hakem A, Wakeham A, Potter J, Reitmair A, Billia F, Firpo E, Hui CC, Roberts J, Rossant J, Mak TW. The tumor suppressor gene *Brca1* is required for embryonic cellular proliferation in the mouse. *Cell* 1996; 85:1009-23.

Ludwig T, Chapman DL, Papaioannou VE, Efstratiadis A. Targeted mutations of breast cancer susceptibility gene homologs in mice: lethal phenotypes of *Brca1*, *Brca2*, *Brca1/ Brca2*, *Brca1/p53* and *Brca2/p53* nullizygous embryos. *Genes Dev* 1997; 11:1226-41.

Gowen LC, Johnson BL, Latour AM, Sulik KK, Koller BH. *Brca1* deficiency results in early embryonic lethality characterized by neuroepithelial abnormalities. *Nat Genet* 1996; 12:191-4.

Shen SX, Weaver Z, Xu X, Li C, Weinstein M, Chen L, Guan XY, Ried T, Deng CX. A targeted disruption of the murine *Brca1* gene causes gamma-irradiation hypersensitivity and genetic instability. *Oncogene* 1998; 17:3115-24.

Xu X, Qiao W, Linke SP, Cao L, Li WM, Furth PA, Harris CC, Deng CX. Genetic interactions between tumor suppressors *Brca1* and *p53* in apoptosis, cell cycle and tumorigenesis. *Nat Genet* 2001; 28:266-71.

Mak TW, Hakem A, McPherson JP, Shehabeldin A, Zablocki E, Migon E, Duncan GS, Bouchard D, Wakeham A, Cheung A, Karaskova J, Sarosi I, Squire J, Marth J, Hakem R. *Brcal* required for T cell lineage development but not TCR loci rearrangement. *Nat Immunol* 2000; 1:77-82.

Xu X, Wagner KU, Larson D, Weaver Z, Li C, Ried T, Hennighausen L, Wynshaw-Boris A, Deng CX. Conditional mutation of *Brca1* in mammary epithelial cells results in blunted ductal morphogenesis and tumour formation. *Nat Genet* 1999; 22:37-43.

Brodie SG, Xu X, Qiao W, Li WM, Cao L, Deng CX. Multiple genetic changes are associated with mammary tumorigenesis in *Brca1* conditional knockout mice. *Oncogene* 2001; 20:7514-23.

Question

Many citations are not directly related to the statements.

Answer

In this 2nd revision, we have checked each reference to ensure their relevance to the related topics. For these less relevant, we either remove or replace them with more relevant references, and new references are also included to related topics as marked in the revision.

Question

The treatment of variants that are considered pathogenic is not reassuring.

Answer

We addressed this issue in our previous revision: "The *BRCA* PLP data used in our study (Supplementary Table 2) was originated from ClinVar. We only used the *BRCA* PLP with single classification as P or LP. These with conflict interpretation, multiple interpretation, and uncertain classification were all excluded from our study to ensure the reliability of the analysis". Among all *BRCA* databases, the *BRCA* variation and classification data from ClinVar is currently the golden standard in guiding clinical interpretation of *BRCA* test results around the world. In terms of quality and quantity, we believe the use of the *BRCA* PLPs from ClinVar is the best assurance to ensure that the *BRCA* variants used in our study are pathogenic and likely pathogenic, and the results really reflect the origins of human *BRCA* PLP, rather than *BRCA*

variants of benign or likely benign or VUS as did by most previous evolution *BRCA* studies (Supplementary table 1)". We consider that the description has addressed the issue.

Question

The premise that the frequency of *BRCA* PLPs are high is, in my view, incorrect, and no evidence is shown to support the numbers. In fact, only in high risk cohorts the prevalence approaches the numbers cited (and implicitly stated to correspond to unselected cohorts).

Answer

The prevalence of *BRCA* PLP is between 0.2-0.5% in modern human population based on others' and our own studies in general populations, or 1 in several hundreds of individuals. In the high-risk cohorts of cancer population is 2-10% in breast cancer (Huszno et al. 2019) and up to 25% in ovarian cancer (Manchana et al. 2019).

In the revision, we included the following description to address the issue:

The prevalence of *BRCA* PLP is between 0.2-0.5% in modern human populations. For example, the prevalence is 0.26% in Japanese population (Momozawa et al. 2018), 0.29% in Macau population (Qin et al. 2021), 0.38% in Chinese population (Dong et al. 2021), 0.38% in Mexican population (Fernandez-Lopez et al. 2019), 0.39% in Malaysian population (Wen et al. 2018), 0.53% in Taiwanese population (Chian et al. 2021), and 0.53% in Caucasian populations (Kurian et al. 2019). That implies that one in several hundreds of human individuals carries a *BRCA* PLP variant. The prevalence of *BRCA* PLP can be the highest in disease-causing genetic predisposition genes in human. It is interesting to understand for why *BRCA* PLP can reach such high level in human population regardless their deleterious impact.

Related references

Huszno J, Kólosza Z, Grzybowska E. *BRCA1* mutation in breast cancer patients: Analysis of prognostic factors and survival. *Oncol Lett.* 2019 Feb;17(2):1986-1995. doi: 10.3892/ol.2018.9770. Epub 2018 Nov 28. PMID: 30675265; PMCID: PMC6341769.

Manchana T, Phoolcharoen N, Tantibirojn P. *BRCA* mutation in high grade epithelial ovarian cancers. *Gynecol Oncol Rep.* 2019 Aug 13;29:102-105. doi: 10.1016/j.gore.2019.07.007. PMID: 31467961; PMCID: PMC6710551.

Wen WX, Allen J, Lai KN, Mariapun S, Hasan SN, Ng PS, et al. (2018). Inherited Mutations in *BRCA1* and *BRCA2* in an Unselected Multiethnic Cohort of Asian Patients with Breast Cancer and Healthy Controls from Malaysia. *J. Med. Genet.* 55 (2), 97–103. doi:10.1136/jmedgenet-2017-104947

Dong, H., Chandratre, K., Qin, Y., Zhang, J., Tian, X., Rong, C., et al. (2020). Prevalence of *BRCA1/BRCA2* Pathogenic Variation in Chinese Han Population. *J. Med. Genet.* [Epub ahead of print]. doi:10.1136/jmedgenet-2020-106970

Qin, Z., Kuok, C. N., Dong, H., Jiang, L., Zhang, L., Guo, M., et al. (2020). Can Population *BRCA* Screening Be Applied in Non-ashkenazi Jewish Populations? Experience in Macau Population. *J. Med. Genet.* [Epub ahead of print]. doi:10.1136/jmedgenet-2020-107181

Momozawa, Y., Iwasaki, Y., Parsons, M. T., Kamatani, Y., Takahashi, A., Tamura, C., et al. (2018). Germline Pathogenic Variants of 11 Breast Cancer Genes in 7,051 Japanese Patients and 11,241 Controls. *Nat. Commun.* 9 (1), 4083. doi:10.1038/s41467-018-06581-8

Kurian, A. W., Ward, K. C., Howlader, N., Deapen, D., Hamilton, A. S., Mariotto, A., et al. (2019). Genetic Testing and Results in a Population-Based Cohort of Breast Cancer Patients and Ovarian Cancer Patients. *Jco* 37, 1305–1315. doi:10.1200/JCO.18.01854

Fernández-Lopez JC, Romero-Córdoba S, Rebollar-Vega R, Alfaro-Ruiz LA, Jiménez- Morales S, Beltrán-Anaya F, Arellano-Llamas R, Cedro-Tanda A, Rios-Romero M, Ramirez-Florencio M, Bautista-Piña V, Dominguez-Reyes C, Villegas-Carlos F, Tenorio- Torres A, Hidalgo-Miranda A. Population and breast cancer patients' analysis reveals the diversity of genomic variation of the BRCA genes in the Mexican population. *Hum Genomics* 2019;13:3.

Question

The approach to compare large population datasets with sequence of homologs as presented seems to lack rigor and the conclusions are not compelling.

Answer

The approaches we used in comparing large population datasets were directly adapted from the human genome project-developed genome conservation system (https://genome.ucsc.edu/cgi-bin/hgTrackUi?hgsid=1261809889_YnIFDiXpOPauYgd69fMAIjflLtAyD&db=hg38&c=chrX&q=cons100way). The details for the system have been well described in many publications, the process is well standardized, and the pipeline has been widely used in various evolution genomic studies. The followings copied from the system show the examples. In response to reviewers' comments, we provided more description for the mapping process in our 1st revision. However, as shown below, the entire system has been well developed. Restricted by the space, there is no need to give a full description for the standardized system. We believe that the data generated by using the system in our study provide high quality. We disagree with reviewer's comments "lack rigor and the conclusions are not compelling".

Reference list from the UCSC conservation system:

References

Phylo-HMMs, phastCons, and phyloP:

Felsenstein J, Churchill GA. A Hidden Markov Model approach to variation among sites in rate of evolution. *Mol Biol Evol*. 1996 Jan;13(1):93-104. PMID: 8583911

Pollard KS, Hubisz MJ, Rosenbloom KR, Siepel A. Detection of nonneutral substitution rates on mammalian phylogenies. *Genome Res*. 2010 Jan;20(1):110-21. PMID: 19858363; PMC: PMC2798823

Siepel A, Bejerano G, Pedersen JS, Hinrichs AS, Hou M, Rosenbloom K, Clawson H, Spieth J, Hillier LW, Richards S, et al. Evolutionarily conserved elements in vertebrate, insect, worm, and yeast genomes. *Genome Res*. 2005 Aug;15(8):1034-50. PMID: 16024819; PMC: PMC1182218

Siepel A, Haussler D. *Phylogenetic Hidden Markov Models*. In: Nielsen R, editor. *Statistical Methods in Molecular Evolution*. New York: Springer; 2005. pp. 325-351. DOI: 10.1007/0-387-27733-1_12

Yang Z. A space-time process model for the evolution of DNA sequences. *Genetics*. 1995 Feb;139(2):993-1005. PMID: 7713447; PMC: PMC1206396

Chain/Net:

Kent WJ, Baertsch R, Hinrichs A, Miller W, Haussler D. Evolution's cauldron: duplication, deletion, and rearrangement in the mouse and human genomes. *Proc Natl Acad Sci U S A*. 2003 Sep 30;100(20):11484-9. PMID: 14500911; PMC: PMC208764

Multiz:

Blanchette M, Kent WJ, Riemer C, Elnitski L, Smit AF, Roskin KM, Baertsch R, Rosenbloom K, Clawson H, Green ED, et al. Aligning multiple genomic sequences with the threaded blockset aligner. *Genome Res*. 2004 Apr;14(4):706-15. PMID: 15060014; PMC: PMC383317

Lastz (formerly Blastz):

Chiaromonte F, Yap VB, Miller W. Scoring pairwise genomic sequence alignments. *Pac Symp Biocomput*. 2002;115-26. PMID: 11928468

Harris RS. *Improved pairwise alignment of genomic DNA*. Ph.D. Thesis. Pennsylvania State University, USA. 2007.

Schwartz S, Kent WJ, Smit A, Zhang Z, Baertsch R, Hardison RC, Haussler D, Miller W. Human-mouse alignments with BLASTZ. *Genome Res*. 2003 Jan;13(1):103-7. PMID: 12529312; PMC: PMC430961

Phylogenetic Tree:

Murphy WJ, Eizirik E, O'Brien SJ, Madsen O, Scally M, Douady CJ, Teeling E, Ryder OA, Stanhope MJ, de Jong WW, Springer MS. Resolution of the early placental mammal radiation using Bayesian phylogenetics. *Science*. 2001 Dec 14;294(5550):2348-51. PMID: 11743200

Description of the methodologies from USCS conservation section:

Methods

Pairwise alignments with the human genome were generated for each species using lastz from repeat-masked genomic sequence. Lineage-specific repeats were removed prior to alignment, then reinserted. Pairwise alignments were then linked into chains using a dynamic programming algorithm that finds maximally scoring chains of gapped subsections of the alignments organized in a kd-tree. The scoring matrix and parameters for pairwise alignment and chaining were tuned for each species based on phylogenetic distance from the reference. High-scoring chains were then phased along the genome, with gaps filled by lower-scoring chains, to produce an alignment net. For more information about the chaining and netting process and parameters for each species, see the description pages for the Chain and Net tracks.

An additional filtering step was introduced in the generation of the 100-way conservation track to reduce the number of paralogous and pseudogenes from the high-quality assemblies and the suspect alignments from the low-quality assemblies: the pairwise alignments of high-quality mammalian sequences (placental and marsupial) were filtered based on synteny; those for 2X mammalian genomes were filtered to retain only alignments of best quality in both the target and query ("reciprocal best").

The resulting best-in-genome pairwise alignments were progressively aligned using multizbaumd, following the tree topology diagrammed above, to produce multiple alignments. The multiple alignments were post-processed to add annotations indicating alignment gaps, genomic breaks, and base quality of the component sequences. The annotated multiple alignments, in IAF format, are available for bulk download. An alignment summary table containing an entry for each alignment block in each species was generated to improve track display performance at large scales. Framing tables were constructed to enable visualization of codons in the multiple alignment display.

Phylogenetic Tree Model

Both phastCons and phyloP are phylogenetic methods that rely on a tree model containing the tree topology, branch lengths representing evolutionary distance at neutrally evolving sites, the background distribution of nucleotides, and a substitution rate matrix. The all-species tree model for this track was generated using the phyloP program from the PHAST package (REV mode, EM algorithm, medium precision) using multiple alignments of 4-45c degenerate sites extracted from the 100-way alignment (msa_vms). The 4c sites were derived from the RefSeq (Reviewed-Coding) gene set, filtered to select single-coverage long transcripts.

This same tree model was used in the phyloP calculations; however, the background frequencies were modified to maintain reversibility. The resulting tree model: all-species.

PhastCons Conservation

The phastCons program computes conservation scores based on a phylo-HMM, a type of probabilistic model that describes both the process of DNA substitution at each site in a genome and the way this process changes from one site to the next (Palamstein and Churchill 1996, Yang 1995, Siepel and Haussler 2005). PhastCons uses a two-state phylo-HMM, with a state for conserved regions and a state for non-conserved regions. The value plotted at each site is the posterior probability that the corresponding alignment column was "generated" by the conserved state of the phylo-HMM. These scores reflect the phylogeny (including branch lengths) of the species in question, a continuous-time Markov model of the nucleotide substitution process, and a tendency for conservation levels to be autocorrelated along the genome (i.e., to be similar at adjacent sites). The general reversible (REV) substitution model was used. Unlike many conservation-scoring programs, phastCons does not rely on a sliding window of fixed size; therefore, short highly-conserved regions and long moderately-conserved regions can both obtain high scores. More information about phastCons can be found in Siepel et al. 2005.

The phastCons parameters used were: expected-length=45, target-coverage=0.3, non=0.3.

PhyloP Conservation

The phyloP program supports several different methods for computing p-values of conservation or acceleration, for individual nucleotides or larger elements (<http://icongen.ucsf.edu/phyloP>). Here it was used to produce separate scores at each base (-wq scores option), considering all branches of the phylogeny rather than a particular subtree or lineage (i.e., the -subtree option was not used). The scores were computed by performing a likelihood ratio test at each alignment column (-method LR7), and scores for both conservation and acceleration were produced (-mode CONACC).

Conserved Elements

The conserved elements were predicted by running phastCons with the -filter option. The predicted elements are segments of the alignment that are likely to have been "generated" by the conserved state of the phylo-HMM. Each element is assigned a log-odds score equal to its log probability under the conserved model minus its log probability under the non-conserved model. The "score" field associated with this track contains transformed log-odds scores, taking values between 0 and 1000. (The scores are transformed using a monotonic function of the form a * log(b + x). The raw log-odds scores are retained in the "name" field and can be seen on the details page or in the browser when the track's display mode is set to "pack" or "full".

Description of 100 vertebrate genome alignment UCSC used for conservation study:

Vertebrate Multiz Alignment & Conservation (100 Species) (*All Comparative Genomics tracks)

Maximum display mode:

Select views (Help):

Multiz Alignments Basewise Conservation (phyloP) Element Conservation (phastCons) Conserved Elements

Multiz Alignments Configuration

Species selection:

Primate

chimp gorilla orangutan gibbon rhesus
 crab-eating macaque baboon green monkey marmoset squirrel monkey
 bushbaby

Euarchontoglires

chinese tree shrew squirrel lesser Egyptian jerboa prairie vole chinese hamster
 golden hamster mouse rat naked mole-rat guinea pig
 chinchilla brush-tailed rat rabbit pika

Laurasiatheria

pig alpaca bactrian camel dolphin killer whale
 tibetan antelope cow sheep domestic goat horse
 white rhinoceros cat dog ferret panda
 pacific walrus weddell seal black flying-fox megabat david's myotis (bat)
 little brown bat big brown bat hedgehog shrew star-nosed mole

Afrotheria

elephant cape elephant shrew manatee cape golden mole tenrec
 aardvark

Mammal

armadillo opossum tasmanian devil wallaby platypus

Birds

saker falcon peregrine falcon collared flycatcher white-throated sparrow medium ground finch
 zebra finch tibetan ground jay budgerigar parrot scarlet macaw
 rock pigeon mallard duck chicken turkey

Sarcopterygii

american alligator green sea turtle painted turtle chinese softshell turtle spiny softshell turtle
 lizard x. tropicalis coelacanth

Fish

tetraodon fugu yellowbelly pufferfish Nile tilapia princess of Burundi
 burton's mouthbreeder zebra mbuna pundamilia nyererei medaka southern platyfish
 stickleback atlantic cod zebrafish mexican tetra (cavefish) spotted gar
 lamprey

Multiple alignment base-level:

Display bases identical to reference as dots
 Display chains between alignments

Codon Translation:

Default species to establish reading frame:

No codon translation
 Use default species reading frames for translation
 Use reading frames for species if available, otherwise no translation
 Use reading frames for species if available, otherwise use default species

Description of Multiz, PhyloP and PhastCons methods UCSC developed for conservation study:

Downloads for data in this track are available:

Top

- Multiz alignments (MAF format), and phylogenetic trees
- PhyloP conservation (WIG format)
- PhastCons conservation (WIG format)

Description

This track shows multiple alignments of 100 vertebrate species and measurements of evolutionary conservation using two methods (*phastCons* and *phyloP*) from the PHAST package, for all species. The multiple alignments were generated using multiz and other tools in the UCSC/Penn State Bioinformatics comparative genomics alignment pipeline. Conserved elements identified by *phastCons* are also displayed in this track. PHAST/Multiz are built from chains ("alignable") and nets ("syntenic"), see the documentation of the Chain/Net tracks for a description of the complete alignment process.

PhastCons is a hidden Markov model-based method that estimates the probability that each nucleotide belongs to a conserved element, based on the multiple alignment. It considers not just each individual alignment column, but also its flanking columns. By contrast, *phyloP* separately measures conservation at individual columns, ignoring the effects of their neighbors. As a consequence, the *phyloP* plots have a less smooth appearance than the *phastCons* plots, with more "texture" at individual sites. The two methods have different strengths and weaknesses. *PhastCons* is sensitive to "runs" of conserved sites, and is therefore effective for picking out conserved elements. *PhyloP*, on the other hand, is more appropriate for evaluating signatures of selection at particular nucleotides or classes of nucleotides (e.g., third codon positions, or first positions of miRNA target sites).

Another important difference is that *phyloP* can measure acceleration (faster evolution than expected under neutral drift) as well as conservation (slower than expected evolution). In the *phyloP* plots, sites predicted to be conserved are assigned positive scores (and shown in blue), while sites predicted to be fast-evolving are assigned negative scores (and shown in red). The absolute values of the scores represent $-\log p$ -values under a null hypothesis of neutral evolution. The *phastCons* scores, by contrast, represent probabilities of negative selection and range between 0 and 1.

Both *phastCons* and *phyloP* treat alignment gaps and unaligned nucleotides as missing data, and both were run with the same parameters.

See also: lastz parameters and other details and chain minimum score and gap parameters used in these alignments.

UCSC has repeatmasked and aligned all genome assemblies, and provides all the sequences for download. For genome assemblies not available in the genome browser, there are alternative assembly hub genome browsers. Missing sequence in any assembly is highlighted in the track display by regions of yellow when zoomed out and by Ns when displayed at base level (see Gap Annotation, below).

Primate subset				
Organism	Species	Release date	UCSC version	Alignment type
Baboon	Papio hamadryas	Mar 2012	Baylor Panu_2.0/papAnu2	Reciprocal best net
Bushbaby	Otolemur garnettii	Mar 2011	Broad/otoGar3	Syntenic net
Chimp	Pan troglodytes	Feb 2011	CSAC 2.1.4/panTro4	Syntenic net
Crab-eating macaque	Macaca fascicularis	Jun 2013	Macaca_fascicularis_5.0/macFas5	Syntenic net
Gibbon	Nomascus leucogenys	Oct 2012	GGSC Nieu3.0/nomLeu3	Syntenic net
Gorilla	Gorilla gorilla gorilla	May 2011	gorGor3.1/gorGor3	Reciprocal best net
Green monkey	Chlorocebus sabaues	Mar 2014	Chlorocebus_sabeus_1.1/chlSab2	Syntenic net
Human	Homo sapiens	Dec 2013	GRCh38/hg38	reference species
Marmoset	Callithrix jacchus	Mar 2009	WUGSC 3.2/calJac3	Syntenic net
Orangutan	Pongo pygmaeus abelii	July 2007	WUGSC 2.0.2/ponAbe2	Reciprocal best net
Rhesus	Macaca mulatta	Oct 2010	BGI CR_1.0/rheMac3	Syntenic net
Squirrel monkey	Saimiri boliviensis	Oct 2011	Broad/saiBol1	Syntenic net
Euarchontoglires subset				
Brush-tailed rat	Octodon degus	Apr 2012	OctDeg1.0/octDeg1	Syntenic net
Chinchilla	Chinchilla lanigera	May 2012	ChiLan1.0/chiLan1	Syntenic net
Chinese hamster	Cricetulus griseus	Jul 2013	C_griseus_v1.0/criGri1	Syntenic net
Chinese tree shrew	Tupaia chinensis	Jan 2013	TupChi_1.0/tupChi1	Syntenic net
Golden hamster	Mesocricetus auratus	Mar 2013	MesAur1.0/mesAur1	Syntenic net
Guinea pig	Cavia porcellus	Feb 2008	Broad/cavPor3	Syntenic net
Lesser Egyptian jerboa	Jaculus jaculus	May 2012	JacJac1.0/jacJac1	Syntenic net
Mouse	Mus musculus	Dec 2011	GRCh38/mm10	Syntenic net
Naked mole-rat	Heterocephalus glaber	Jan 2012	Broad/HetGla_female_1.0/hetGla2	Syntenic net
Pika	Ochotona princeps	May 2012	OchPri3.0/ochPri3	Syntenic net
Prairie vole	Microtus ochrogaster	Oct 2012	MicOch1.0/micOch1	Syntenic net
Rabbit	Oryctolagus cuniculus	Apr 2009	Broad/oryCun2	Syntenic net
Rat	Rattus norvegicus	Jul 2014	RGSC 6.0/m6	Syntenic net
Squirrel	Spermophilus tridecemlineatus	Nov 2011	Broad/speTri2	Syntenic net
Laurasiatheria subset				
Alpaca	Vicugna pacos	Mar 2013	Vicugna_pacos-2.0.1/vicPac2	Syntenic net
Bactrian camel	Camelus ferus	Dec 2011	CB1/camFer1	Syntenic net
Big brown bat	Eptesicus fuscus	Jul 2012	EptFus1.0/eptFus1	Syntenic net
Black flying-fox	Pteropus alecto	Aug 2012	ASM32557v1/pteAle1	Syntenic net
Cat	Felis catus	Nov 2014	ICGSC Felis_catus_8.0/felCat8	Syntenic net
Cow	Bos taurus	Jun 2014	Bos_taurus_UMD_3.1.1/bosTau8	Syntenic net

Question

The authors continue to make broad-stroke conclusions with references that are clearly inappropriate.

Answer

In the revision, we have revised the tone for the conclusion to be more conserved to avoid overstate the data:

Abstract

.....Based on the data, we conclude that human *BRCA* pathogenic variants are **very likely** arisen in recent human history after the latest out-of-Africa migration, and the expansion of modern human population **could largely increase** the spectrum of variation.

Introduction

.....Data from our study provide evidence to show that human *BRCA* PLP **was highly unlikely** originated from cross-species evolution conservation but **most likely** arisen by positive selection in the humans, during recent human history after migrating out-of-Africa and expansion of human population.

Discussion

..... **Based on these observations, we propose that human *BRCA* PLP was most likely arisen in recent human history, possibly within a few thousand years, after the latest human out-of-Africa migration and settlement at different global destinations. We consider that the positive selection on human *BRCA* could play a major role, and the population expansion could further increase the prevalence of human *BRCA* PLP in modern human population (Figure 4).**

Question

The passage referring to the Tasmanian Devil transmissible face tumor (which has been well characterized and is a unrelated tumor) with a reference from thirty years before the cloning of *BRCA1*, is just one example.

Answer

Fully agree. It has been determined that the disease is related with the **fusion of chromosome 1 and X**, but not related with *BRCA* mutations (Taylor et al, 2017). We have added three more recent references for the issue:

In the revision, the section has been revised as the following:

For example, Tasmanian devil shared 20 human *BRCA* PLP variants. Tasmanian has high risk of developing facial cancer, which is related with the fusion of chromosome 1 and X (Hawkins et al, 2006; Murchison et al, 2010; Bender et al. 2014; Taylor et al. 2017), but no evidence to show that the cancer is related with the *BRCA* PLP variants shared with human.

New references

Hawkins, Clare & Baars, C. & Hesterman, Heather & Hocking, Greg & Jones, M.E. & Lazenby, Billie & Mann, D. & Mooney, N. & Pemberton, David & Pyecroft, Stephen & Restani, M. & Wiersma, J.. (2006). Emerging disease and population decline of an island endemic, the Tasmanian devil *Sarcophilus harrisii*. *Biological Conservation*. 131. 307-324. 10.1016/j.biocon.2006.04.010.

Taylor RL, Zhang Y, Schöning JP, Deakin JE. Identification of candidate genes for devil facial tumour disease tumorigenesis. *Sci Rep*. 2017 Aug 18;7(1):8761. doi: 10.1038/s41598-017-08908-9. PMID: 28821767; PMCID: PMC5562891.

Murchison EP, Tovar C, Hsu A, Bender HS, Kheradpour P, Rebbeck CA, Obendorf D, Conlan C, Bahlo M, Blizzard CA, Pyecroft S, Kreiss A, Kellis M, Stark A, Harkins TT, Marshall Graves JA, Woods GM, Hannon GJ, Papenfuss AT. The Tasmanian devil transcriptome reveals Schwann cell origins of a clonally transmissible cancer. *Science*. 2010 Jan 1;327(5961):84-7. doi: 10.1126/science.1180616. PMID: 20044575; PMCID: PMC2982769.

January 21, 2022

RE: Life Science Alliance Manuscript #LSA-2021-01263-TRR

Prof. San Ming Wang
University of Macau
Faculty of Health Sciences
University of Macau
Taipa
Macau SAR, Taipa 999078
Macao

Dear Dr. Ming Wang,

Thank you for submitting your revised manuscript entitled "Human BRCA pathogenic variants were originated during recent human history". We would be happy to publish your paper in Life Science Alliance pending final revisions necessary to meet our formatting guidelines.

- please add the Twitter handle of your host institute/organization as well as your own or/and one of the authors in our system
- Please upload all figure files as individual ones, including the main figure files; all figure legends should only appear in the main manuscript file
- please add your main, supplementary figure, and table legends to the main manuscript text after the references section
- please add callouts for Figure S3A, B to your main manuscript text

A. FINAL FILES:

B. MANUSCRIPT ORGANIZATION AND FORMATTING:

Sincerely,

January 26, 2022

RE: Life Science Alliance Manuscript #LSA-2021-01263-TRRR

Prof. San Ming Wang
University of Macau
Faculty of Health Sciences
University of Macau
Taipa
Macau SAR, Taipa 999078
Macao

Dear Dr. Wang,

Thank you for submitting your Research Article entitled "Human BRCA pathogenic variants were originated during recent human history". It is a pleasure to let you know that your manuscript is now accepted for publication in Life Science Alliance. Congratulations on this interesting work.

DISTRIBUTION OF MATERIALS:

Again, congratulations on a very nice paper. I hope you found the review process to be constructive and are pleased with how the manuscript was handled editorially. We look forward to future exciting submissions from your lab.

Sincerely,
